# Leveraging Recursive Gumbel-Max Trick for Approximate Inference in Combinatorial Spaces

**Kirill Struminsky**[*]
HSE University
Moscow, Russia
k.struminsky@gmail.com

**Artyom Gadetsky**[*†]
HSE University
Moscow, Russia
artygadetsky@yandex.ru

**Denis Rakitin**[*]
HSE University, Skoltech[‡]
Moscow, Russia
rakitindenis32@gmail.com

**Danil Karpushkin**
AIRI,[§] Sber AI Lab, MIPT[¶]
Moscow, Russia
kardanil@mail.ru

**Dmitry Vetrov**
HSE University, AIRI[§]
Moscow, Russia
vetrovd@yandex.ru

## Abstract

Structured latent variables allow incorporating meaningful prior knowledge into deep learning models. However, learning with such variables remains challenging because of their discrete nature. Nowadays, the standard learning approach is to define a latent variable as a perturbed algorithm output and to use a differentiable surrogate for training. In general, the surrogate puts additional constraints on the model and inevitably leads to biased gradients. To alleviate these shortcomings, we extend the Gumbel-Max trick to define distributions over structured domains. We avoid the differentiable surrogates by leveraging the score function estimators for optimization. In particular, we highlight a family of recursive algorithms with a common feature we call stochastic invariant. The feature allows us to construct reliable gradient estimates and control variates without additional constraints on the model. In our experiments, we consider various structured latent variable models and achieve results competitive with relaxation-based counterparts.

## 1 Introduction

To this day, the majority of deep learning architectures consists of differentiable computation blocks and relies on gradient estimates for learning. At the same time, architectures with discrete intermediate components are a good fit for incorporating inductive biases [3, 48] or dynamic control flow [26, 12]. One of the approaches to train such architectures is to replace the discrete component with a stochastic latent variable and optimize the expected objective.

In practice, the expectation has high computational cost, thus one typically resorts to stochastic estimates for the expectation and its gradient. Particularly, the two prevalent approaches to estimate the gradient of the objective are the score function estimator [45] and the reparameterization trick [16, 40] for relaxed discrete variables [29, 15]. The former puts mild assumptions on the distribution and the objective, requiring the gradient of log-probability with respect to the distribution parameters to be differentiable, and provides unbiased estimates for the objective gradient. However, the naive

---

[*]Equal contribution
[†]Corresponding author
[‡]Skolkovo Institute of Science and Technology
[§]Artificial Intelligence Research Institute
[¶]Moscow Institute of Physics and Technology

35th Conference on Neural Information Processing Systems (NeurIPS 2021).

estimate suffers from high variance and is less intuitive in implementation. In comparison, the reparameterized gradient estimates seamlessly integrate within the backpropagation algorithm and exhibit low variance out of the box. At the same time, the relaxation requires an architecture to be defined on the extended domain of the relaxed variable and introduces bias to the gradient estimate.

In the recent years, the attention of the community shifted towards models with structured latent variables. Informally, a structured variable models a distribution over structured objects such as graphs [4, 34], sequences [9] or matchings [31]. Such latent variable may alter the computation graph or represent a generative process of data. Often, a structured variable is represented as a sequence of categorical random variables with a joint distribution incorporating the structure constraints (e.g., the fixed number of edges in an adjacency matrix of a tree). Recent works on structured latent variables address model training largely through the reparameterization trick using relaxed variables. In fact, the Gumbel-Softmax trick naturally translates to structured variables when $\arg\max$ operator is applied over a structured domain rather than component-wise [34]. In contrast, score function estimators are now less common in structured domain, with a few exceptions such as [50, 14]. The primary difficulty is the sample score function: neither Gibbs distributions, nor distribution defined through a generative process have a general shortcut to compute it.

In our work, we develop a framework to define structured variables along with a low-variance score function estimator. Our goal is to allow training models that do not admit relaxed variables and to improve optimization by alleviating the bias of the relaxed estimators. To achieve the goal we define the structured variable as an output of an algorithm with a perturbed input. Then, we outline a family of algorithms with a common property we call *stochastic invariant*. The property was inspired by the observation in [34, Appendix, Sec. B], where the authors showed that the Kruskal's algorithm [24] and the CLE algorithm [7] are recursively applying the Gumbel-Max trick. We construct new algorithms with the same property and show how to use the property to learn structured latent variables. In the experimental section, we report performance on par with relaxation-based methods and apply the framework in a setup that does not allow relaxations.

## 2 The Recursive Gumbel-Max Trick in Algorithms With Stochastic Invariants

The section below shows how to define a distribution over structured domain. Conceptually, we define a structured random variable as an output of an algorithm with a random input (e.g., to generate a random tree we return the minimum spanning tree of a graph with random weights). A common solution to incorporate such variable in a latent variable model is to replace the original algorithm with a differentiable approximation to allow gradient-based learning[4, 31]. Such solution bypasses the difficulty of computing the sample probability. In contrast, we outline a family of algorithms for which we can get the probability of each intermediate computation step. To get the probabilities we restrict our attention to algorithms with specific recursive structure and random inputs with exponential distribution. In the next section, we leverage the probabilities for approximate inference in latent variable models without the differentiable approximations of the algorithm.

### 2.1 The Gumbel-Max Trick in $\arg\mathrm{top}\,k$

We illustrate our framework with a recursive algorithm generating a subset of a fixed size. The lemma below is a well-known result used to generate categorical random variables using a sequence of exponential random variables.

**Lemma 1.** *(the Exponential-Min trick) If $E_i \sim \mathrm{Exp}\,(\lambda_i), i \in \{1, \ldots, d\}$ are independent, then for $X := \mathrm{argmin}_i E_i$*

1. *the outcome probability is $\mathbb{P}_X(X = x; \lambda) \propto \lambda_x$;*

2. *random variables $E_i' := E_i - E_X, i \in \{1, \ldots, d\}$ are mutually independent given $X$ with $E_i' \mid X \sim \mathrm{Exp}(\lambda_i)$ when $i \neq X$ and $E_i' = 0$ otherwise.*[6]

The lemma is equivalent to *The Gumbel-Max trick*, defined for the variables $G_i := -\log E_i, i \in \{1, \ldots, d\}$ and the maximum position $\mathrm{argmax}_i G_i$. In the above case, a variable $G_i$ has a Gumbel

---

[6]As a convention, we assume that $0 \stackrel{d}{=} \mathrm{Exp}\,(\infty)$.

distribution with the location parameter $\theta_i = \log \lambda_i$, hence the name. Though the Gumbel-Max trick formulation is more common in the literature, we formulate the framework in terms of the exponential distribution and the equivalent Exponential-Min trick. Although the two statements are equivalent and we use their names interchangeably, some of the examples have a natural formulation in terms of the exponential distribution.

Importantly, the second claim in Lemma 1 allows applying the Exponential-Min trick succesively. We illustrate this idea with an algorithm for finding top-k elements. We present the recursive form of $\arg\operatorname{top} k$ in Algorithm 1. For each recursion level, the algorithm finds the minimum element, decrements $k$ and calls itself to find the subset excluding the minimum variable. For reasons explained below, the algorithm subtracts the minimum from the sequence $E'_j = E_j - E_T$ before the recursion. This step does not change the output and may seem redundant.

Assuming the input of the algorithm is a vector $E$ of independent exponential variables with rate parameters $\lambda$, the first argument in the recursive call $E'$ is again a vector of independent exponential variables (given $T$) due to Lemma 1. In other words, the input distribution class is *invariant* throughout the recursion. Subtraction of the minimum is not necessary, but it allows to apply Lemma 1 directly and simplifies the analysis of the algorithms. Besides that, for each recursion level variable $T$ has categorical distribution (conditioned on $T$ found in the above calls) with output probabilities proportional to $\lambda_k, k \in K$.

We use upper indices to denote the recursion depth and, with a slight abuse of notation, denote the concatenation of variables $T$ for each recursion depth as $T := (T^1, \ldots, T^k)$. The output $X$ is a set and does not take into account the order in $T$. Intuitively, $T$ acts as *the execution trace* of the algorithm, whereas $X$ contains only partial information about $T$. The marginal probability of $x$ is the sum $\mathbb{P}_X(X = x; \lambda) = \sum_{t \in X^{-1}(x)} \mathbb{P}_T(T = t; \lambda)$ over all possible orderings of $x = \{x_1, \ldots, x_k\}$ denoted as $X^{-1}(x)$. The direct computation of such sum is prohibitive even for moderate $k$.

The $\arg\operatorname{top} k$ illustration is a well-known extension of the Exponential-Min trick. In particular, the distribution of $T$ is known as the Plackett-Luce distribution [37] and coincides with $k$ categorical samples without replacement. Following the recursion, the observation probability factorizes according to the chain rule with $i$-th factor governed by equation $\mathbb{P}_{T_i}(T_i = t_i \mid t_1, \ldots, t_{i-1}; \lambda) = \frac{\lambda_{t_i}}{\sum_{j=1}^d \lambda_j - \sum_{j=1}^{i-1} \lambda_{t_j}}$. We discuss the multiple applications of the trick in Section 4. Next, we extend Algorithm 1 beyond subset selection.

## 2.2 General Algorithm With the Stochastic Invariant

In this section, we generalize Algorithm 1. The idea is to preserve the property of Algorithm 1 that allows applying the Exponential-Min trick and abstract away the details to allow various instantiations of the algorithm. Algorithm 2 is the generalization we put next to Algorithm 1 for comparison. It has a similar recursive structure and abstracts away the details using the auxiliary subrouties: $f_{\text{stop}}$ is the stop condition, $f_{\text{map}}$ and $f_{\text{combine}}$ handle the recursion and $f_{\text{split}}$ is an optional subroutine for the Exponential-Min trick. Additionally, we replace $k$ with an auxiliary argument $R$ used to accumulate information from the above recursion calls. Below, we motivate the subroutines and discuss the properties of a arbitrary instance of Algorithm 2.

After checking the stop condition with $f_{\text{stop}}$, Algorithm 2 applies the Exponential-Min trick simultaneously over $m$ disjoint sets rather than the whole index set $K$. For example, such operation occurs when we find columnwise minimum in CLE algorithm[7]. To allow the operation we construct *a partition* of indices $P_1, \ldots, P_m$ and find the $\arg\min$ across the partition sets. To generate the partition, we introduce a new subroutine $f_{split}$ taking the index set $K$ and the auxiliary argument $R$ as inputs. The partition size $m$ may also be variable.

After the $m$ simultaneous Exponential-Min tricks, the generalized algorithm calls $f_{\text{map}}$ to select a subset of indices $K' \subsetneq K$ and to accumulate the necessary information for the next call in $R'$. Intuitively, the argument $R'$ represents *a reduction* to a smaller problem solved with a recursive call. In the $\arg\operatorname{top} k$ example, $K'$ is $K \setminus \{T\}$ and $R'$ is the decrement $k - 1$. Note that Algorithm 2 does not allow to capture such information with the other inputs $E'$ and $K'$ exclusively.

**Algorithm 1** $F_{\text{top-k}}(E, K, k)$ - finds $k$ smallest elements in a sequence $E$, where $K$ is the set of indices *(keys)* of $E$

**Input:** $E, K, k$
**Output:** $X$
  **if** $k = 0$ **then**
    **return**
  **end if**
  {Find the smallest element}
  $T \Leftarrow \arg\min_{j \in K} E_j$
  **for** $j \in K$ **do**
    $E'_j \Leftarrow E_j - E_T$
  **end for**
  {Exclude $\arg\min$ index $T$, decrement $k$}
  $K', k' \Leftarrow K \setminus \{T\}, k - 1$
  $E' \Leftarrow \{E'_k \mid k \in K'\}$
  {Solve for $k' = k - 1$}
  $X' \Leftarrow F_{\text{top-k}}(E', K', k')$
  **return** $\{T\} \cup X'$

**Algorithm 2** $F_{\text{struct}}(E, K, R)$ - returns structured variable $X$ based on utilities $E$ indexed by $K$ and an auxiliary variable $R$

**Input:** $E, K, R$
**Output:** $X$
  **if** $f_{\text{stop}}(K, R)$ **then**
    **return**
  **end if**
  $P_1, \ldots, P_m \Leftarrow f_{\text{split}}(K, R)$    $\{\sqcup_{i=1}^m P_i = K\}$
  **for** $i = 1$ to $m$ **do**
    $T_i \Leftarrow \arg\min_{j \in P_i} E_j$
    **for** $j \in P_i$ **do**
      $E'_j \Leftarrow E_j - E_{T_i}$
    **end for**
  **end for**
  $K', R' \Leftarrow f_{\text{map}}(K, R, \{T_i\}_{i=1}^m)$   $\{K' \subsetneq K\}$
  $E' \Leftarrow \{E'_k \mid k \in K'\}$
  $X' \Leftarrow F_{\text{struct}}(E', K', R')$    {Recursive call}
  **return** $f_{\text{combine}}(X', K, R, \{T_i\}_{i=1}^m)$

Figure 1: The recursive algorithm for $\arg\text{top } k$ and the general algorithm with the stochastic invariant put side-by-side. Both algorithm perform the Exponential-Min trick and proceed with recursion using a subset of variables. The output $X$ combines the current trace $T$ and the recursion output $X'$.

Finally, the algorithm calls $f_{\text{combine}}$ to construct the structured variable $X$ using the recursive call output $X'$ and the other variables. In the top-k example, $f_{\text{combine}}$ appends the minimum variable index $T$ to the set $X'$.

Now we argue that Algorithm 2 preserves the invariant observed in Algorithm 1. Again, we call the sequence of variables $T = (T_1, \ldots, T_m)$ *the trace* of the algorithm. By design, if the input $E$ is a sequence of independent exponential random variables, then the recursion input $E'$ conditioned on $T$ is again a sequence of independent exponential distributions. For short, we call this property *the stochastic invariant*. The key to the stochastic invariant is the signature of the subroutines Algorithm 2 uses. The algorithm only accesses $E$ values though the Exponential-Min trick. As a result, the intermediate variables $K'$ and $R'$ as well as the output $X$ depend on $E$ only through $T$. In other words, the execution trace is a function of perturbation $T = T(E)$ and the structured variable $X = X(T)$ is a function of the trace. Additionally, due to Lemma 1, the trace components $T_1, \ldots, T_m$ have categorical distributions, whereas $E'_k, k \in K$ are exponential random variables. We prove these properties by induction w.r.t. the recursion depth in Appendix A.

Given the above, we derive two modifications of Algorithm 2 generalizing Lemma 1 and the Plackett-Luce distribution from the illustration. Algorithm 3 computes the log-probability $\log \mathbb{P}_T(t; \lambda)$ of a trace realization $t$. In Section 3, we use the algorithm output to construct gradient estimators. Again, the pseudo-code introduces index $j$ to denote the recursion depth and assumes the input $t = \{t_i^j\}_{i,j}$ is the concatenation of trace variables for all recursion depths $j = 1, \ldots, k$. Similarly, in Appendix B we present an algorithm returning a sample from $E \mid T = t$ given trace realization $t$.

## 2.3 Further Examples

This subsection contains an overview of algorithms with stochastic invariants along with the corresponding structured variables. We present the details and the pseudo-code in Appendix E.

Analogous to the $\arg\text{top } k$ and the subset variable, the insertion sorting algorithm is an algorithm with the stochastic invariant. In the case of sorting, we do not omit the order of the trace variable $T$ and return the permutation $X = T$. The resulting variable $X$ has the Plackett-Luce distribution. We use the variable as a latent variable for insertion-based non-monotonic generation [13]. As an alternative to the Plackett-Luce distribution, we consider a square parameter matrix and find a matching between rows and columns. We perturb the matrix and iteratively find the minimum element

**Algorithm 3** $F_{\text{log-prob}}(t, \lambda, K, R)$ - returns $\log \mathbb{P}_T(t; \lambda)$ for trace $t$, rates $\lambda$, $K$ and $R$ as in Alg. 2

---

**Input:** $t, \lambda, K, R$
**Output:** $\log \mathbb{P}_T(t; \lambda)$
  **if** $f_{\text{stop}}(K, R)$ **then**
    **return**
  **end if**
  $P_1, \ldots, P_m \Leftarrow f_{\text{split}}(K, R)$
  **for** $i = 1$ to $m$ **do**
    $\log \mathbb{P}_T(t_i^1; \lambda) \Leftarrow \log \lambda_{t_i^1} - \log\left(\sum_{k \in P_i} \lambda_k\right)$         {Index $j$ in $T_i^j$ denotes the recursion level}
    **for** $k \in P_i \setminus \{t_i^1\}$ **do**
      $\lambda'_k \Leftarrow \lambda_k$
    **end for**
    $\lambda'_{t_i^1} \Leftarrow +\infty$                                               {Because $E'(t_i^1) = 0$}
  **end for**
  $K', R' \Leftarrow f_{\text{map}}(K, R, \{t_i^1\}_{i=1}^m)$
  $\lambda' \Leftarrow \{\lambda'_k \mid k \in K'\}$
  $\log \mathbb{P}_T(t^{>1} \mid T^1 = t^1; \lambda) \Leftarrow F_{\text{log-prob}}(t^{>1}, \lambda', K', R')$    {Compute log-prob of $t^{>1} := \{t_i^j\}_{j>1}$}
  **return** $\sum_{i=1}^m \log \mathbb{P}_T(t_i^1; \lambda) + \log \mathbb{P}_T(t^{>1} \mid T^1 = t^1; \lambda)$

---

in the matrix. We exclude the row and the column containing the element and proceed to the next recursion step. Notably, in contrast to this algorithm, the Hungarian algorithm [25] for the minimum weight matching does not have the stochastic invariant.

As [34] observe, Kruskal's algorithm [24] and Chu-Liu-Edmonds [7] algorithm recursively apply the Exponential-Min trick, thus have the stochastic invariant. The former constructs the minimum spanning (MST) tree edge-by-edge. The corresponding trace variable $T$ is a sequence of edges, whereas $X$ is an unordered set of edges. Interestingly, we could not represent Prim's algorithm [38, 5] for the MST as an algorithm with the stochastic invariant. The Chu-Liu-Edmonds algorithm is an analog of Kruskal's algorithm for directed graphs. It returns the minimum tree $X$ with a fixed root.

Additionally, we construct a latent binary tree variable specifically for the non-monotonic generation model [44]. In this model, each token in the sentence corresponds to a node in the tree. We assign weights to nodes and perturb the weights to obtain a sample. During the recursion, we set the minimum weight node to be the parent. We put the nodes on the left-hand side to the left subtree and the nodes on the right-hand side to the right subtree.

## 3 Gradient Estimation for the Recursive Gumbel-Max Trick

In this section, we develop the gradient estimates for the structured distributions defined in Subsection 2.2. We start with a brief introduction of the two main approaches to gradient estimation for discrete categorical variables: the score function estimator [45] and the Gumbel-Softmax estimator [29, 15]. Then, we propose a low-variance modification of the score function estimator for the structured variables based on the intermediate representation of the variable. Finally, we conclude with a discussion of control variates we use together with the proposed estimator.

### 3.1 Gradient Estimation for Categorical Variables

We consider gradient estimates of an expected objective $\nabla_\lambda \mathbb{E}_X \mathcal{L}(X)$, where a discrete random variable $X$ has parametric distribution $\mathbb{P}_X(\cdot; \lambda)$ with finite support. The basic *score function estimator* [45], also known as REINFORCE, defines an unbiased estimate for the gradient using a sample $x$ as $\mathcal{L}(x) \nabla_\lambda \log \mathbb{P}_X(X = x; \lambda)$. The estimator does not make any assumptions about $\mathcal{L}(\cdot)$, but requires an efficient sampling algorithm for $X$ and the score function $\nabla_\lambda \log \mathbb{P}_X(X = x; \lambda)$. For a categorical random variable $X$ with outcome probabilities $\mathbb{P}_X(X = k; \lambda) \propto \lambda_k$ computation of $\nabla_\lambda \log \mathbb{P}_X(X = x; \lambda)$ is linear in the number of outcomes $d$. Therefore, the gradient estimation is fast when $d$ is small. However, for structured variables, such as graphs or sequences, the number of outcomes $d$ grows rapidly with the structure size. In this case, the estimator requires custom algorithms for sampling and estimating the score function.

*The Gumbel-softmax estimator*, introduced in [15, 29], is an alternative estimator that defines a continuous relaxation based on Lemma 1. On the forward pass, it replaces the categorical variable $X$ with a differentiable surrogate $\tilde{x} = \operatorname{soft max}\left(\frac{g}{\tau}\right)$, where the input $g := (-\log e_1, \ldots, -\log e_d)$ is a component-wise transformation of exponential samples. Due to Lemma 1, the surrogate converges to the one-hot encoding of a categorical sample $x$ as $\operatorname{soft max}$ converges to $\arg\max$ when $\tau \to 0$. On the backward pass, the estimator uses the chain rule to construct the gradient $\nabla_\lambda \mathcal{L}(\tilde{x})$ using the reparameterization trick [16, 40] to define the partial derivative of a sample as $\frac{\partial e_i}{\partial \lambda_i} = -\frac{e_i}{\lambda_i}$. The Gumbel-Softmax estimator naturally extends to structured variables [31, 34]. Specifically, the component-wise optimization in Lemma 1 can be replaced with a linear program over a structured set to generate structured variables and a relaxation can be used to define gradients. In the experimental section, we consider Stochastic Softmax Tricks (SST), introduced in [34], as the relaxation-based baseline for comparison.

As opposed to the score function estimator, the Gumbel-Softmax estimator requires a differentiable loss $\mathcal{L}(\cdot)$. Such requirement imposes an additional restriction on a model architecture. The architecture must be defined for the relaxed samples as well as the hard samples, a non-trivial requirement for the models where discrete variables define branching [26] or the inputs the model is not allowed to see [19, 13]. In practice, the vanilla score function estimator has notoriously higher variance compared to the Gumbel-Softmax estimator and requires a control variate to improve the gradient descent convergence.

## 3.2 The Score Function Estimator for the Recursive Gumbel-Max Trick

In Subsection 2.2, we have introduced a probabilistic model involving an exponential variable $E$ and the structured variable $X$ defined as an output of an algorithm with input $E$. Additionally, we have defined an intermediate trace variable $T = T(E)$ such that $X$ is a function $X = X(T)$. In this subsection, we apply $T$ to estimate gradients of $\mathbb{E}_X \mathcal{L}(X)$.

In our setup, the score function $\nabla_\lambda \log \mathbb{P}_E(E = e; \lambda)$ is available out of the box. However, the score function estimator

$$g_E := \mathcal{L}(X(e))\nabla_\lambda \log \mathbb{P}_E(E = e; \lambda), \tag{1}$$

which we refer to as $E$-REINFORCE, is rarely used in practice. In fact, the variance of the score function estimator using $E$ as an action space exceeds the variance of the estimator using $X$. On the other hand, the score function for the structured variable $\nabla_\lambda \log \mathbb{P}_X(X = x; \lambda)$ involves marginalization w.r.t. $E$ and may require significant computation resources to estimate.

To mitigate the variance of $g_E$, we follow the observation of [43, Appendix, Sec. B] and define another estimator as

$$g_T := \mathcal{L}(X(t))\nabla_\lambda \log \mathbb{P}_T(T = t; \lambda), \tag{2}$$

which is essentially the score function estimator that uses $T$ rather than $E$. Below we refer to it as $T$-REINFORCE. Such estimate can be seen as the score function estimator $g_E$ marginalized over $E$ given $T$ (Appendix A contains the detailed derivation)

$$\mathcal{L}(X)\nabla_\lambda \log \mathbb{P}_T(T; \lambda) = \mathbb{E}_{E|T}\left[\mathcal{L}(X)\nabla_\lambda \log \mathbb{P}_E(E; \lambda) \mid T\right]. \tag{3}$$

As a result, the proposed gradient estimate $g_T$ is unbiased

$$\mathbb{E}_E \mathcal{L}(X)\nabla_\lambda \log \mathbb{P}_E(E; \lambda) = \mathbb{E}_T \mathbb{E}_{E|T}\left[\mathcal{L}(X)\nabla_\lambda \log \mathbb{P}_E(E; \lambda) \mid T\right], \tag{4}$$

whereas the variance of the estimate does not exceed the variance of $g_E$

$$\operatorname{var}_E[g_E] = \operatorname{var}_T\left[\mathbb{E}_{E|T} g_E\right] + \mathbb{E}_T \operatorname{var}_{E|T}[g_E] = \operatorname{var}_T[g_T] + \mathbb{E}_T \operatorname{var}_{E|T}[g_E] \geq \operatorname{var}_T[g_T]. \tag{5}$$

In fact, in our experiments, we observed a significant difference in optimization due to the reduced variance of the estimator.

As we have argued in Subsection 2.2, we can compute the score function for trace variable and apply the estimator $g_T$ in practice. Similarly, marginalization with respect to $T \mid X$ leads to the score function estimator $g_X := \mathcal{L}(x)\nabla_\lambda \log \mathbb{P}_X(X = x; \lambda)$ and reduces the variance even further $g_T : \operatorname{var}_T[g_T] \geq \operatorname{var}_X[g_X]$. Therefore, the standard score function estimator is preferable when $\nabla_\lambda \log \mathbb{P}_X(X = x; \lambda)$ is available. In other cases, $g_T$ is a practical alternative.

### 3.3 Further Variance Reduction for the Score Function Estimator

In addition to the marginalization described above, we mitigate the variance of the score function estimator with control variates. We use two strategies to construct the control variates. The first strategy uses the algorithm for conditional reparameterization of $E \mid T$ (Appendix B, Algorithm 6) and defines a family of sample-dependent control variates for the score function estimator [43, 11]. The estimator generates a sample $e$, runs the corresponding algorithm to obtain $t$ and $x = X(t)$, adds a control variate $c(e)$ and uses an independent sample $\tilde{e}$ from the conditional distribution $E \mid T = t$ to eliminate the introduced bias

$$\left(\mathcal{L}(X(t)) - c(\tilde{e})\right) \nabla_\lambda \log \mathbb{P}_T(T = t; \lambda) - \nabla_\lambda c(\tilde{e}) + \nabla_\lambda c(e). \tag{6}$$

In general, the above estimate extends to any pair of random variables $(B, Z)$ such that $B = B(Z)$ and the conditional distribution $Z \mid B$ admits the reparameterization trick. In [43], the control variate used the relaxed loss $\mathcal{L}(\cdot)$, whereas [11] proposed to learn the control variate to improve the training dynamic. In our experiments, we use the estimator of [11] and refer to it as RELAX.

The second family of control variates we consider uses $K > 1$ samples $t_1, \dots, t_K$ to reduce the variance. Besides averaging the independent estimates, it uses the objective sample mean $\bar{\mathcal{L}} := \frac{\sum_{i=1}^K \mathcal{L}(X(t_i))}{K}$ to reduce the variance even further:

$$\frac{1}{K-1} \sum_{i=1}^K \left(\mathcal{L}(X(t_i)) - \overline{\mathcal{L}}\right) \nabla_\lambda \log \mathbb{P}_T(T = t_i; \lambda). \tag{7}$$

Despite being quite simple, the above leave-one-out estimator [20] proved to be competitive with multiple recent works [6, 41]. In our experiments, we refer to such estimator as $T$-REINFORCE+.[7] To facilitate fair comparison, in a batch training setup we reduce the batch size proportionally to $K$.

## 4 Related Work

Early models with structured latent variables include HMMs [39], PCFGs [36], make strong assumptions about the model structure, and typically use EM-algorithm variations for training. This paper continues the line of work on perturbation models [33] for distributions over combinatorial sets. Initially, perturbation models approximated Gibbs distributions with an efficient sampling procedure using the MAP oracle for the Gibbs distribution. Later, [15, 29] proposed to relax the component-wise optimization used in the Gumbel-Max trick to facilitate gradient-based learning for Gibbs distributions. The combination of the two approaches, namely a perturbed model together with a bespoke relaxation of the MAP oracle, allows designing learning algorithms for latent subsets [46], permutations [31], trees [4] and sequences [9]. Recently, [34] developed a systematic approach to design relaxations for perturbed models with linear MAP oracle. Unlike the previous works, we mainly focus on the score function estimators [45] for learning.

We illustrate the framework with the well-known Gumbel-Top-k trick [49]. Among the various applications, [47] used the trick to define a differentiable relaxation for the subset model considered in our paper; meanwhile, [10, 42] used the trick to define score function estimators for latent permutations. Besides that, [21, 22] leveraged the trick for sampling without replacement for a certain family of graphical models and design a gradient estimator using the sampler [20]. Importantly, [34, Appendix, Sec. B] showed that the Kruskal's algorithm and the Chu-Liu-Edmonds algorithm extend the Gumbel-Top-k trick. They used the observation to argue in favor of exponential perturbations. In turn, we generalize the observation and propose a learning algorithm based on the generalization.

The conditional reparameterization scheme, proposed in our work, allows action-dependent control variates [43, 11] for learning the structured variables. However, similarly to [41, 6] we observed that often a simpler leave-one-out baseline [20] has better performance. Besides the control variates, [35] recently adopted conditional reparameterization to construct an improved Gumbel Straight-Through estimator [15] for the categorical latent variables.

---

[7]We denote the analogue, which uses the exponential score instead of the score of the trace, by $E$-REINFORCE+

Table 1: Results of $k$-subset selection on Aroma aspect data. MSE ($\times 10^{-2}$) and subset precision (%) is shown for best models selected on validation averaged across different random seeds.

| MODEL | ESTIMATOR | $k = 5$ | | $k = 10$ | | $k = 15$ | |
|---|---|---|---|---|---|---|---|
| | | MEAN $\pm$ STD | PREC. | MEAN $\pm$ STD | PREC. | MEAN $\pm$ STD | PREC. |
| SIMPLE | SST (Our Impl.) | $\mathbf{3.6 \pm 0.1}$ | $28 \pm 1.4$ | $3.21 \pm 0.12$ | $29.5 \pm 1.7$ | $2.77 \pm 0.09$ | $\mathbf{28.1 \pm 1.2}$ |
| | E-REINFORCE+ | $3.89 \pm 0.2$ | $25 \pm 1.4$ | $3.77 \pm 0.23$ | $26.7 \pm 3.4$ | $3.16 \pm 0.16$ | $25.3 \pm 1.3$ |
| | T-REINFORCE+ | $3.79 \pm 0.13$ | $\mathbf{30.5 \pm 2.2}$ | $\mathbf{3.14 \pm 0.16}$ | $\mathbf{31 \pm 2.9}$ | $\mathbf{2.69 \pm 0.11}$ | $27.6 \pm 0.9$ |
| | RELAX | $3.76 \pm 0.11$ | $24 \pm 1.9$ | $3.5 \pm 0.13$ | $28.9 \pm 1.9$ | $2.95 \pm 0.15$ | $26.1 \pm 1.9$ |
| COMPLEX | SST (Our Impl.) | $2.93 \pm 0.09$ | $\mathbf{56 \pm 2.1}$ | $2.55 \pm 0.08$ | $\mathbf{49.4 \pm 3.1}$ | $2.51 \pm 0.05$ | $40.3 \pm 0.9$ |
| | E-REINFORCE+ | $3.03 \pm 0.2$ | $49.4 \pm 3.3$ | $2.92 \pm 0.12$ | $45 \pm 2.3$ | $2.76 \pm 0.22$ | $42 \pm 2.2$ |
| | T-REINFORCE+ | $\mathbf{2.75 \pm 0.08}$ | $55.8 \pm 3.4$ | $\mathbf{2.48 \pm 0.05}$ | $48.6 \pm 2.4$ | $\mathbf{2.4 \pm 0.03}$ | $\mathbf{44.2 \pm 1.3}$ |
| | RELAX | $2.8 \pm 0.08$ | $54.4 \pm 2.1$ | $2.58 \pm 0.09$ | $47.6 \pm 1.6$ | $2.46 \pm 0.08$ | $42.1 \pm 1.5$ |

## 5 Applications

In the section below, we study various instances of algorithms with stochastic invariants and compare them against relaxation-based Stochastic Softmax Tricks (SST) introduced in [34]. SST offers a generalization of the well-known Gumbel-Softmax trick to the case of different structured random variables. The experimental setup is largely inherited from [34], apart from Subsection 5.4 where the specifics of the problem do not allow relaxation-based gradient estimators, thus showing broader applicability of the score function-based gradient estimators. The main goal of [34] was to show that introducing structure will lead to superior performance compared to unstructured baselines. In turn, we focus on studying the benefits that one could get from using score-function gradient estimators based on $T$ rather than $E$ as well as showing competitive performance compared to relaxation-based SSTs.

Concerning the efficiency of the proposed score-function-based gradient estimators, the recursive form of Algorithm 2 does not facilitate batch parallelization in general. Specifically, the recursion depth and the decrease in the size of $E$ may differ within a batch. Therefore, Algorithms 2,3 may require further optimization. We discuss our implementations in Appendix, Section C, and provide speed benchmarks in Table 9. It shows that, in practice, the performance is not much inferior to the relaxation-based competitor. The implementation and how-to-use examples are publicly available[8].

### 5.1 Learning to Explain by Finding a Fixed Size Subset

We evaluated our method on the experimental setup from L2X [2] on the BeerAdvocate [30] dataset. The dataset consists of textual beer reviews and numerical ratings for four beer aspects (*Aroma*, *Taste*, *Appearance*, *Palate*). The model utilizes encoder-decoder architecture. The encoder outputs parameters of top-$k$ distribution over subsets of a review given the entire review. The decoder predicts a rating given the review subset of size $k$. Following setup of [34] we use $k = \{5, 10, 15\}$ and two CNN architectures: *Simple* — a one-layer CNN and *Complex* — a three-layer CNN and train our models for each aspect using MSE loss. We train models using several score function based estimators and compare them with our implementation of SST. We report means and standard deviations for loss and precision averaged across different random model initializations. Table 1 shows the obtained results for *Aroma* aspect. The best metrics with respect to means are highlighted in bold. Detailed experimental setup and description of models can be found in Appendix D.1.

### 5.2 Learning Latent Spanning Trees with Kruskal's Algorithm

Given a system of interacting particles, the dependencies of their states can be formally described as an undirected graph. We use Neural Relational Inference [17], initially representing a relaxation-based approach, and examine its performance as a generative model and ability to reconstruct system interconnections, when applying score function techniques instead. We build an experiment in line with [34], generating data by translating a ground truth latent spanning tree (corresponding to the connections in the system) into a sequence of positions of points on a real plane, representing

---

[8]https://github.com/RakitinDen/pytorch-recursive-gumbel-max-trick

Table 2: Graph Layout experiment results for T=10 iterations. Metrics are obtained by choosing models with best validation ELBO and averaging results across different random seeds on the test set.

| | $T = 10$ | | | | | |
|---|---|---|---|---|---|---|
| ESTIMATOR | ELBO | | EDGE PREC. | | EDGE REC. | |
| | MEAN $\pm$ STD | MAX | MEAN $\pm$ STD | MAX | MEAN $\pm$ STD | MAX |
| *SST (Our Impl.)* | $-1860.79 \pm 1116.83$ | $-1374.81$ | $\mathbf{87 \pm 21}$ | $\mathbf{95}$ | $\mathbf{93 \pm 3}$ | $\mathbf{95}$ |
| *T-REINFORCE+* | $\mathbf{-1582.72 \pm 571.18}$ | $\mathbf{-1192.04}$ | $70 \pm 31$ | $91$ | $86 \pm 8$ | $91$ |
| *RELAX* | $-2079.18 \pm 569.49$ | $-1205.87$ | $43 \pm 31$ | $90$ | $81 \pm 6$ | $90$ |

Table 3: Unsupervised Parsing on ListOps. We report the average test-performance of the model with the best validation accuracy across different random initializations.

| ESTIMATOR | ACCURACY | | PRECISION | | RECALL | |
|---|---|---|---|---|---|---|
| | MEAN $\pm$ STD | MAX | MEAN $\pm$ STD | MAX | MEAN $\pm$ STD | MAX |
| *SST (Our Impl.)* | $78.42 \pm 8.14$ | $\mathbf{93.78}$ | $56.84 \pm 20.08$ | $\mathbf{82.40}$ | $30.18 \pm 19.10$ | $73.11$ |
| *E-REINFORCE+* | $60.25 \pm 2.29$ | $64.47$ | $40.87 \pm 6.90$ | $45.74$ | $40.74 \pm 6.93$ | $45.46$ |
| *T-REINFORCE+* | $\mathbf{87.34 \pm 3.00}$ | $91.97$ | $\mathbf{77.93 \pm 7.36}$ | $79.65$ | $\mathbf{61.10 \pm 14.11}$ | $\mathbf{79.65}$ |
| *RELAX* | $79.60 \pm 9.36$ | $88.64$ | $54.73 \pm 17.48$ | $75.27$ | $53.61 \pm 17.14$ | $75.27$ |

dynamics of the system over time. These points are obtained executing force-directed algorithm [8] for $T = 10$ or $T = 20$ iterations and fed into the model.

The architecture of the model consists of a graph neural network (GNN) encoder, producing distribution over spanning trees, and a GNN decoder, producing distribution over time series of points positions. Model is trained in a manner of variational autoencoders (VAEs), optimizing ELBO, a lower bound on the joint log-likelihood of the observed data points at all timesteps.

We measure precision and recall with respect to the encoder samples and the ground truth dependency spanning tree. Table 2 shows the results for T=10 iterations. Overall, score function methods performed better than their relaxation-based counterpart, achieving higher values of ELBO on the test set, but slightly worse performance in terms of structure recovery metrics. The results for T=20 and the detailed experimental setup are described in the Appendix D.2.

### 5.3 Unsupervised Parsing with Rooted Trees with CLE Algorithm

We study the ability of the proposed score function estimators to recover the latent structure of the data in a setting, where it can be quite accurately described with an arborescence. Following details about data and models outlined by [34], we use a simplified version of the ListOps [32] dataset. It consists of mathematical expressions (e.g. min[3 med[3 5 4] 2]), written in prefix form along with results of their evaluation, which are integers in $[0, 9]$. Given a prefix expression, one can algorithmically recover its structure as a parse tree. We bound maximal length of expressions and maximal depth of their parses along with removing the examples with summod operator. These limitations sufficiently decrease the amount of memory a model should have to calculate the result and facilitates the usage of GNNs which now become capable of evaluating expressions by a bounded number of message passing steps.

Our model consists of two parts: an encoder and a classifier. The encoder is a pair of LSTMs that generate parameters of the distribution over rooted arborescence on token nodes. The classifier is a GNN, which passes a fixed number of messages over the sampled arborescence and feeds the resulting embedding of the first token into the final MLP. Models are trained simultaneously to minimize cross-entropy. We examine the performance of the models by measuring classification accuracy along with precision and recall with respect to the edges of ground truth parse trees. Table 3 shows score function based estimators, particularly T-REINFORCE+, show more stable performance in comparison to relaxation-based estimator. Detailed description of the experiment can be found in the Appendix D.3.

### 5.4 Non-monotonic Generation of Balanced Parentheses with Binary Trees

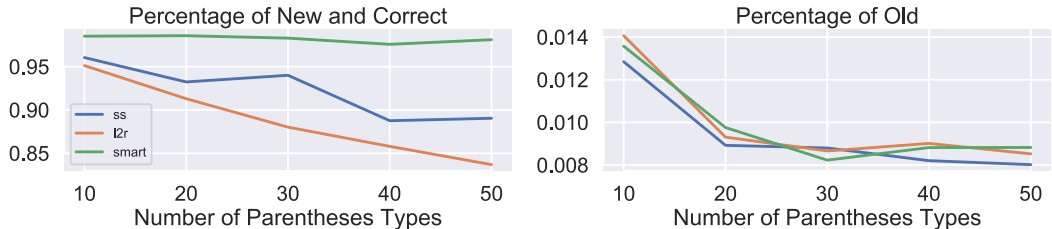

Figure 2: Generative statistics of non-monotonic language model with different orders

We apply our methods in a setting where the application of continuous relaxations is seemingly not possible. We construct a simple dataset by sampling a fixed number of balanced parentheses of various types (from 10 to 50 types) and model them with non-monotonic architecture, defined in [44]. Here, a language model generates sequences by consecutively adding new tokens in between previously generated ones. This process can be seen as modeling joint distribution over binary trees (treated as orders of generation) and sequences (which are generated along these orders).

We refer to this probabilistic model as the decoder, fix its architecture to be a single-layer LSTM and use teacher-forcing for training. More precisely, we compare two different modes of training decoder: fixed order and semi-supervised trainable order. In the first case, the order is obtained by applying a deterministic function to the input sequence. We study two fixed orders: *left-to-right*, corresponding to a degenerate tree with only right child nodes, and a more natural *smart* order, described in Appendix D.4. In the semi-supervised (*ss*) case, 10% of sequences are paired with the smart order, and the overall model is trained as a VAE. We train the decoder by directly backpropagating the output signal and obtain the gradients for the encoder using RELAX estimator.

We choose models with the best validation perplexity and evaluate them by generating 200000 unique sequences and measuring portion of those which are present in the train dataset (Old) and those which are balanced and not seen during training (New and Correct). Results of this toy experiment show that it is possible to improve generative metrics by considering non-trivial orders of generation. Experiment is described in details in the Appendix D.4.

## 6 Discussion

Below, we speculate about the pros and cons of the relaxation-based [34] and the score function-based approaches to training latent structured variable models. While both build upon the Gumbel-Max trick, the generalizations develop different ideas. Our work summarizes the recent observations and formulates a general algorithm for which the Gumbel-Max applies recursively. We utilize the properties of these algorithms to construct unbiased score function-based gradient estimators. In contrast, [34] generalizes the relaxation of the Gumbel-Max trick to different combinatorial structures and produces biased reparametrization-based gradient estimators. While the relaxation-based gradient estimators are biased and limited to differentiable objectives, they have lower variance and faster batch processing time (Tables 8 and 9). At the same time, score function-based estimators are unbiased and apply when the loss functions do not admit relaxations (Section 5.4). Occasionally, they lead to better optimization and lower objective values (Tables 10, 11 and 12). The choice of control variates also introduces a trade-off in our framework. As the experiments show, if the objective is highly parallelable, the multi-sample $T$-REINFORCE+ estimator is preferable. However, a single-sample RELAX estimator is a more suitable choice when we cannot obtain multiple objective samples (i.e., in reinforcement learning). Finally, as a direction for future improvements, we suggest applying the conditional reparameterization scheme used in the control variate to improve the Gumbel straight-through estimators.

**Acknowledgements**

The authors thank the reviewers for the valuable feedback. The research was supported by the Russian Science Foundation grant no. 19-71-30020 and through the computational resources of HPC facilities at HSE University[23].

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
