## Appendix Outline

## A  Proofs

In this section, we provide the proofs for the lemmas and the theorem formulated in the main part.

### A.1  The Exponential-Min Trick

**Lemma 2.** *(the Exponential-Min trick) If $E_i \sim \mathrm{Exp}(\lambda_i), i \in \{1, \ldots, d\}$ are independent, then for $X := \mathrm{argmin}_i E_i$*

*1. the outcome probability is $\mathbb{P}_X(X = x; \lambda) \propto \lambda_x$;*

*2. random variables $E_i' := E_i - E_X, i \in \{1, \ldots, d\}$ are mutually independent given $X$ with $E_i' \mid X \sim \mathrm{Exp}(\lambda_i)$ when $i \neq X$ and $E_i' = 0$ otherwise.[9]*

*Proof.* Starting with the joint density of $X$ realization $x$ and $E$ realization $e$

$$\prod_{j=1}^{d} \left( \lambda_j e^{-\lambda_j e_j} \cdot \mathbb{I}(e_x \leq e_j) \right), \tag{8}$$

we make the substitution $e_j' := e_j - e_x$ for $j \in \{1, \ldots, d\}$ and rewrite the density

$$\prod_{j=1}^{d} \left( \lambda_j e^{-\lambda_j (e_j' + e_x)} \cdot \mathbb{I}(0 \leq e_j') \right) = \lambda_x e^{-\sum_{j=1}^{d} \lambda_j e_x} \prod_{j=1}^{d} \left( \lambda_j e^{-\lambda_j e_j'} \mathbb{I}(0 \leq e_j') \right) = \tag{9}$$

$$\frac{\lambda_x}{\sum_{j=1}^{d} \lambda_j} \times \left( \sum_{j=1}^{d} \lambda_j \right) e^{-(\sum_{j=1}^{d} \lambda_j) e_x} \times \prod_{j=1}^{d} \left( \lambda_j e^{-\lambda_j e_j'} \mathbb{I}(0 \leq e_j') \right). \tag{10}$$

The latter is the joint density of

- a categorical latent variable $X$ with $\mathbb{P}(X = x) \propto \lambda_x$;

- an independent exponential random variable $E_X$ with rate parameter $\sum_{j=1}^{d} \lambda_j$;

- and a sequence of random variables $E_j' := E_j - E_X, j \neq x$ with mutually independent exponential distributions $\mathrm{Exp}(\lambda_j)$ conditioned on $X = x$.

$\square$

### A.2  Properties of the Score Function Estimators

This subsection contains the analysis of mean and variance of the defined score function estimators with respect to variables $E, T$ and $X$. We follow the derivations in [43, Appendix, Sec. B] and start with lemma about the conditional marginalization. We assume applicability of the log-derivative trick every time it is used.

---

[9]As a convention, we assume that $0 \overset{d}{=} \mathrm{Exp}(\infty)$.

**Lemma 3.** *Consider a random variable $Y$ with distribution from parametric family $\mathbb{P}_Y(\cdot; \lambda)$ and a function $Z = Z(Y)$. Then, $Z$-REINFORCE estimator is the marginalization of $Y$-REINFORCE estimator with respect to the distribution $Y \mid Z$:*

$$\mathbb{E}_{Y|Z}\left[\mathcal{L}(Z)\nabla_\lambda \log \mathbb{P}_Y(Y; \lambda) \mid Z\right] = \mathcal{L}(Z)\nabla_\lambda \log \mathbb{P}_Z(Z; \lambda). \tag{11}$$

*Proof.* Since $\mathcal{L}(Z)$ is a function of $Z$, it can be moved outside of the conditional expectation:

$$\mathbb{E}_{Y|Z}\left[\mathcal{L}(Z)\nabla_\lambda \log \mathbb{P}_Y(Y; \lambda) \mid Z\right] = \mathcal{L}(Z)\mathbb{E}_{Y|Z}\left[\nabla_\lambda \log \mathbb{P}_Y(Y; \lambda) \mid Z\right]. \tag{12}$$

The only thing that remains is to show that the $Y$-score marginalized over $Y \mid Z$ equals $Z$-score:

$$\mathbb{E}_{Y|Z}\left[\nabla_\lambda \log \mathbb{P}_Y(Y; \lambda) \mid Z\right] = \nabla_\lambda \log \mathbb{P}_Z(Z; \lambda). \tag{13}$$

We start by rewriting the log-probability of $Y$ as the difference between the joint and the conditional log-probabilities:

$$\mathbb{E}_{Y|Z}\left[\nabla_\lambda \log \mathbb{P}_Y(Y; \lambda) \mid Z\right] = \mathbb{E}_{Y|Z}\left[\nabla_\lambda \log \mathbb{P}_{Y,Z}(Y, Z; \lambda) - \nabla_\lambda \log \mathbb{P}_{Z|Y}(Z \mid Y; \lambda) \mid Z\right]. \tag{14}$$

Next, we observe that the conditional log-probability $\log \mathbb{P}_{Z|Y}(Z \mid Y; \lambda)$ is concentrated in $Z = Z(Y)$ and equals $\log \mathbb{I}(Z = Z(Y))$, which is zero for all $Y \sim \mathbb{P}_{Y|Z}(\cdot \mid Z; \lambda)$. Thus, the second summand cancels out:

$$\mathbb{E}_{Y|Z}\left[\nabla_\lambda \log \mathbb{P}_Y(Y; \lambda) \mid Z\right] = \mathbb{E}_{Y|Z}\left[\nabla_\lambda \log \mathbb{P}_{Y,Z}(Y, Z; \lambda) \mid Z\right]. \tag{15}$$

Then we rewrite the joint score as the sum of marginal $Z$-score and the conditional score of $Y$ given $Z$:

$$\mathbb{E}_{Y|Z}\left[\nabla_\lambda \log \mathbb{P}_{Y,Z}(Y, Z; \lambda) \mid Z\right] = \mathbb{E}_{Y|Z}\left[\nabla_\lambda \log \mathbb{P}_Z(Z; \lambda) + \nabla_\lambda \log \mathbb{P}_{Y|Z}(Y \mid Z; \lambda)\right]. \tag{16}$$

Finally, the expectation of the score with respect to the corresponding distribution is zero:[10]

$$\mathbb{E}_{Y|Z}\left[\nabla_\lambda \log \mathbb{P}_{Y|Z}(Y \mid Z; \lambda) \mid Z\right] = \nabla_\lambda \mathbb{E}_{Y|Z}\left[1 \mid Z\right] = 0. \tag{17}$$

At the same time, $Z$-score is already the function of $Z$ and can be moved outside of expectation:

$$\mathbb{E}_{Y|Z}\left[\nabla_\lambda \log \mathbb{P}_{Y,Z}(Y, Z; \lambda) \mid Z\right] = \mathbb{E}_{Y|Z}\left[\nabla_\lambda \log \mathbb{P}_Z(Z; \lambda) \mid Z\right] + 0 = \nabla_\lambda \log \mathbb{P}_Z(Z; \lambda). \tag{18}$$

Combining all previous steps, we arrive at:

$$\mathbb{E}_{Y|Z}\left[\mathcal{L}(Z)\nabla_\lambda \log \mathbb{P}_Y(Y; \lambda) \mid Z\right] = \mathcal{L}(Z)\nabla_\lambda \log \mathbb{P}_Z(Z; \lambda). \tag{19}$$

$\square$

The fact above also generalizes to the case of REINFORCE with baselines:

**Corollary 1.** *Let $Y$ and $Z$ be random variables defined as in Lemma 3 and $C$ be a random variable, independent of $Y$. Then, $Z$-REINFORCE with baseline $C$ is the marginalization of $Y$-REINFORCE with baseline $C$ with respect to the distribution $Y \mid Z, C$:*

$$\mathbb{E}_{Y|Z,C}\left[\left(\mathcal{L}(Z) - C\right)\nabla_\lambda \log \mathbb{P}_Y(Y; \lambda) \mid Z, C\right] = \left(\mathcal{L}(Z) - C\right)\nabla_\lambda \log \mathbb{P}_Z(Z; \lambda). \tag{20}$$

*Proof.* Start by observing that the loss part $(\mathcal{L}(Z) - C)$ is a function of $Z$ and $C$; it can be moved outside of the conditional expectation:

$$\mathbb{E}_{Y|Z,C}\left[\left(\mathcal{L}(Z) - C\right)\nabla_\lambda \log \mathbb{P}_Y(Y; \lambda) \mid Z, C\right] = \left(\mathcal{L}(Z) - C\right)\mathbb{E}_{Y|Z,C}\left[\nabla_\lambda \log \mathbb{P}_Y(Y; \lambda) \mid Z, C\right]. \tag{21}$$

---

[10]Here we apply the log-derivative trick to $\log \mathbb{P}_{Y|Z}(Y \mid Z; \lambda)$. While the trick may not apply to an arbitrary distribution, it is easy to show that the trick is correct for the conditional distributions we consider in our work.

Next, since $Y$ and $C$ are independent and $Z$ is a deterministic function of $Y$, the whole random vector $(Y, Z)$ is independent from $C$. Given this, one can remove the conditioning on $C$:

$$(\mathcal{L}(Z) - C) \, \mathbb{E}_{Y|Z,C} \left[ \nabla_\lambda \log \mathbb{P}_Y(Y; \lambda) \mid Z, C \right] = (\mathcal{L}(Z) - C) \, \mathbb{E}_{Y|Z} \left[ \nabla_\lambda \log \mathbb{P}_Y(Y; \lambda) \mid Z \right]. \quad (22)$$

Finally, Lemma 3 states that the remaining conditional expectation is just the $Z$-score:

$$(\mathcal{L}(Z) - C) \, \mathbb{E}_{Y|Z} \left[ \nabla_\lambda \log \mathbb{P}_Y(Y; \lambda) \mid Z \right] = (\mathcal{L}(Z) - C) \, \nabla_\lambda \log \mathbb{P}_Z(Z; \lambda). \quad (23)$$

$\square$

To draw the connection between the above statements and the estimators, defined in Section 3, we observe that the execution trace variable $T$ is the deterministic function of algorithm's input, exponential random vector $E$, and the execution trace is defined in such a way, that the discrete random variable $X$ can be recovered from it, i.e. $X$ is a deterministic function of $T$, which is true for the algorithms we take into consideration (see Theorem 1). Given this, we formulate the following lemma:

**Lemma 4.** *Let $E$, $T$ and $X$ be random variables defined as in section 3, i.e. $E$ is the exponential random vector with (multidimensional) parameter $\lambda$, $T = T(E)$ is the function of $E$ (execution trace) and $X = X(T)$ is the function of $T$ (structured variable). For each of the variables define the corresponding REINFORCE estimator with baseline random variable $C$, independent of $E$:*

$$g_E = (\mathcal{L}(X) - C) \, \nabla_\lambda \log \mathbb{P}_E(E; \lambda), \quad (24)$$

$$g_T = (\mathcal{L}(X) - C) \, \nabla_\lambda \log \mathbb{P}_T(T; \lambda), \quad (25)$$

$$g_X = (\mathcal{L}(X) - C) \, \nabla_\lambda \log \mathbb{P}_X(X; \lambda). \quad (26)$$

*Then, all the defined gradients are unbiased estimates of the true gradient:*

$$\mathbb{E}_{E,C}[g_E] = \mathbb{E}_{T,C}[g_T] = \mathbb{E}_{X,C}[g_X] = \nabla_\lambda \mathbb{E}_X \mathcal{L}(X). \quad (27)$$

*With the following inequality between their variances:*

$$\mathrm{var}_{X,C}\left[g_X\right] \leq \mathrm{var}_{T,C}\left[g_T\right] \leq \mathrm{var}_{E,C}\left[g_E\right]. \quad (28)$$

*Proof.* We first observe that using change of variables theorem for Lebesgue integral (particularly, law of the unconscious statistician) one can rewrite the true gradient using expectation with respect to $E$ and then apply the log-derivative trick:

$$\nabla_\lambda \mathbb{E}_X \mathcal{L}(X) = \nabla_\lambda \mathbb{E}_E \mathcal{L}(X) = \mathbb{E}_E \mathcal{L}(X) \nabla_\lambda \log \mathbb{P}_E(E; \lambda). \quad (29)$$

Since the baseline $C$ is independent from $E$, subtracting it from the loss does not change the expectation:

$$\mathbb{E}_{E,C} C \cdot \nabla_\lambda \log \mathbb{P}_E(E; \lambda) = \mathbb{E}_C C \cdot \mathbb{E}_E \nabla_\lambda \log \mathbb{P}_E(E; \lambda) = \mathbb{E}_C C \cdot \nabla_\lambda \mathbb{E}_E 1 = 0. \quad (30)$$

Thus, gradient estimator with respect to $E$ is unbiased:

$$\mathbb{E}_{E,C}[g_E] = \mathbb{E}_{E,C} \left(\mathcal{L}(X) - C\right) \nabla_\lambda \log \mathbb{P}_E(E; \lambda) = \mathbb{E}_E \mathcal{L}(X) \nabla_\lambda \log \mathbb{P}_E(E; \lambda) = \nabla_\lambda \mathbb{E}_X \mathcal{L}(X). \quad (31)$$

Next, we use the fact that $T$ is the function of $E$ and $X$ is the function of $T$. Corollary 1 then states that the estimates $g_T$ and $g_X$ can be obtained by applying two sequential conditional marginalizations:

$$g_T = \mathbb{E}_{E|T,C}\left[g_E \mid T, C\right], \quad g_X = \mathbb{E}_{T|X,C}\left[g_T \mid X, C\right]. \quad (32)$$

Since the conditional expectation preserves the expectation, the above two gradients are also unbiased:

$$\mathbb{E}_{T,C}\left[g_T\right] = \mathbb{E}_{X,C}\left[g_X\right] = \nabla_\lambda \mathbb{E}_X \mathcal{L}(X) \quad (33)$$

Finally, conditional expectation reduces variance. For the trace random variable:

$$\mathrm{var}_{E,C}\left[g_E\right] = \mathbb{E}_{T,C} \, \mathrm{var}_{E|T,C}\left[g_E \mid T, C\right] + \mathrm{var}_{T,C}\left[\mathbb{E}_{E|T,C}\left[g_E \mid T, C\right]\right] =$$
$$= \mathbb{E}_{T,C} \, \mathrm{var}_{E|T,C}\left[g_E \mid T, C\right] + \mathrm{var}_{T,C}\left[g_T\right]$$

Observing that the conditional variance $\mathrm{var}_{E|T,C}\left[g_E \mid T, C\right]$ is non-negative, we get:

$$\mathrm{var}_{E,C}\left[g_E\right] \geq \mathrm{var}_{T,C}\left[g_T\right]. \quad (34)$$

Applying the same reasoning for $g_X$, we obtain:

$$\mathrm{var}_{T,C}\left[g_T\right] \geq \mathrm{var}_{X,C}\left[g_X\right]. \quad (35)$$

$\square$

### A.3   Distributions of $T$ and $E \mid T$

Below we prove the main claim of this work. To simplify the argument, we will introduce an additional notation to denote the variables on various recursion depths. We will assume that the first level of recursion has index $j = 1$. We will denote the input variables as $E^1, K^1, R^1$ and as $E, K, R$ interchangeably.

Similarly, we will add the depth index to the partition, the trace variables and the updated exponential variables

$$P_1^j, \ldots, P_{m_j}^j := f_{\text{split}}(K^j, R^j) \tag{36}$$

$$T_i^j := \arg\min_{k \in P_i^j} E_k^j \tag{37}$$

$$E_k^{j+1} := \begin{cases} E_k^j - E_{T_{i(k,j)}^j}^j, & \text{if } k \in K^j \\ E_k^j, & \text{otherwise,} \end{cases} \tag{38}$$

where the function $i(k, j)$ returns the index of the partition set containing key $k$ on depth $j$. To update the index of the auxiliary variables $K$ and $R$ we use the $f_{\text{map}}(\cdot)$ subroutine:

$$K^{j+1}, R^{j+1} := f_{\text{map}}(K^j, R^j, \{T_i^j\}_{i=1}^{m_j}). \tag{39}$$

Finally, we will use $\lambda_k^j$ to denote the updated parameter $\lambda_k^1 := \lambda_k$ after transformations at the depth $j - 1$. It is equal to the $\lambda_k$, if $k$ was not argminimum at all recursion depths from 1 to $j - 1$ or is equal to $+\infty$ otherwise. Given the above notation we formulate

**Theorem 1.** *Let $E$ be a set of exponential random variables $E_k \sim \text{Exp}(\lambda_k)$ indexed by $k \in K$, let $T$ be the trace and $X$ be the output of Algorithm 2 respectively. If the subroutine $f_{stop}$ is defined in such a way that $f_{stop}(\emptyset, R) = 1$ for all possible auxiliary variables $R$, then:*

1. *The output $X$ is determined by the trace $T$ and the auxiliary variables $K, R$: $X(E, T, K, R) = X(T, K, R)$;*[11]

2. *The trace $T$ is a sequence of categorical latent variables $T_i^j$. Each $T_i^j$ is defined at the recursion level $j$ and is either deterministic or has (conditional) outcome probabilities proportional to $\lambda_k$ for $k \in P_i^j$;*

3. *The elements of the conditional distribution $E \mid T$ are distributed as a sum of exponential random variables*

$$E_k \mid T \sim \sum_{j=1}^{N(k)} \text{Exp}\left( \sum_{k' \in P_{i(k,j)}^j} \lambda_{k'}^j \right) + \text{Exp}\left( \lambda_k^{N(k)+1} \right), \tag{40}$$

*where $N(k)$ is the deepest recursion level $j$ such that the index $k$ is contained in the index set $K^j$.*

*Proof.* First, we note that the recursion depth of Algorithm 2 is limited. Indeed, when we make a recursive call, the second argument of $F_{\text{struct}}$ is the index set $K'$, which, by construction, is a strict subset of the finite index set $K$ from the previous call of the function. It means that either $f_{\text{stop}}(K, R)$ is true at some point, or there exists a stage where we call $F_{\text{struct}}$ with $K = \emptyset$. By the assumption, the stop condition is true for the empty set $f_{\text{stop}}(\emptyset, R) = 1$, which means that we always reach the last level of recursion. Next, we will prove the first claim of the theorem by induction on the recursion depth.

We start by considering an input $E^1, K^1, R^1$ for which the algorithm has recursion depth $N = 1$. For a single recursion layer Algorithm 2 checks the stop condition and halts. The stop condition is a function of $K^1$ and $R^1$, therefore the output does not depend on $E$. When an input leads to an arbitrary recursion depth $N$, the algorithm returns $f_{\text{combine}}(X^2, K^1, R^1, T^1)$. We will argue that all of the arguments of the function depend on $E^1$ only through $T$, therefore the algorithm output can be represented as a function of $T$ rather than $E^1$.

---

[11]In the paper we omit the dependence on $K$ and $R$ since these are typically fixed.

By the induction hypothesis, the intermediate structure $X^2 = F_{\text{struct}}(E^2, K^2, R^2)$ is determined by $T^{>1}$, $K^2$ and $R^2$. To obtain $K^2, R^2 = f_{\text{map}}(K^1, R^1, T^1)$ we do not take $E^1$ as an input, therefore $X^2$ is determied by $T^1$. The arguments $K^1$ and $R^1$ of $f_{\text{combine}}$ also do not depend on $E^1$. The last argument $T^1$ depends on $E^1$, but for a given $T^1$ the variability in $E^1$ does not change the output of $f_{\text{combine}}$. Therefore, $X^1$ is determined by the trace $T$ and the auxiliary inputs $K^1$ and $R^1$.

To prove the second and the third claim, we repeat the derivation of Lemma 2 for the joint density of $E$ and $T$. We denote the realizations of $T$ and $E$ with lower-case letters. Here, variables of the form $e_k^j$ are defined analogously to $E_k^j$, but for the corresponding realizations. Recall that $T = \{T_i^j\}_{i,j}$ is the concatenation of trace variables $T_1^j, \ldots, T_{m_j}^j$ for all recursion depths $j = 1, \ldots, k$. We regroup the joint density using the partition $P_1^1, \ldots P_{m_1}^1$ of $K = \sqcup P_i^1$ and splitting the indicator, corresponding to the conditional distribution $T \mid E$, into two indicators, corresponding to the first level of recursion and to the remaining ones respectively:

$$\mathbb{I}(t = T(e)) \prod_{k \in K} \text{Exp}\left(e_k \mid \lambda_k\right) = \mathbb{I}(t = T(e)) \prod_{k \in K} \lambda_k e^{-\lambda_k e_k} \tag{41}$$

$$= \mathbb{I}(t^{>1} = T^{>1}(e)) \cdot \mathbb{I}(t^1 = T^1(e)) \cdot \prod_{k \in K} \lambda_k e^{-\lambda_k e_k} = \tag{42}$$

$$= \mathbb{I}(t^{>1} = T^{>1}(e)) \cdot \prod_{i=1}^{m_1} \prod_{k \in P_i^1} \lambda_k e^{-\lambda_k e_k} \mathbb{I}(e_{t_i^1} \leq e_k). \tag{43}$$

Then we apply Lemma 2 for each $i = 1, \ldots, m_1$ and rewrite the internal product $\prod_{k \in P_i^1}$ as

$$\frac{\lambda_{t_i^1}}{\sum_{k \in P_i^1} \lambda_k} \cdot \left(\sum_{k \in P_i^1} \lambda_k\right) e^{-\left(\sum_{k \in P_i} \lambda_k\right) e_{t_i^1}} \cdot \prod_{k \in P_i^1} \left(\lambda_k^2 e^{-\lambda_k^2 e_k^2} \mathbb{I}(0 \leq e_k^2)\right), \tag{44}$$

where $e_k^2$ is the realization of the random variable $E_k^2 := E_k - E_{T_i^1}$ and $\lambda_k^2 := \lambda_k$ for $k \neq t_i^1$ and $\lambda_{t_i^1}^2 := +\infty$.

To improve readability, we rewrite the same product as

$$\frac{\lambda_{t_i^1}}{\sum_{k \in P_i^1} \lambda_k} \cdot \text{Exp}\left(e_{t_i^1} \mid \sum_{k \in P_i^1} \lambda_k\right) \cdot \prod_{k \in P_i^1} \text{Exp}\left(e_k^2 \mid \lambda_k^2\right) \tag{45}$$

and substitute it into the overall joint density:

$$\mathbb{I}(t^{>1} = T^{>1}(e)) \cdot \prod_{i=1}^{m_1} \frac{\lambda_{t_i^1}}{\sum_{k \in P_i^1} \lambda_k} \cdot \text{Exp}\left(e_{t_i^1} \mid \sum_{k \in P_i^1} \lambda_k\right) \cdot \prod_{k \in P_i^1} \text{Exp}\left(e_k^2 \mid \lambda_k^2\right). \tag{46}$$

Since the execution trace variables at depths $> 1$ are determined by the transformed values $e^2 = \{e_k^2 \mid k \in K\}$, the indicator can be rewritten as $\mathbb{I}(t^{>1} = T^{>1}(e^2))$. At the same time, $\prod_{i=1}^{m_1} \prod_{k \in P_i^1} \text{Exp}\left(e_k^2 \mid \lambda_k^2\right)$ can be rewritten as just $\prod_{k \in K} \text{Exp}\left(e_k^2 \mid \lambda_k^2\right)$, since $P_1^1, \ldots P_{m_1}^1$ is the partition of $K$. Given this, we rewrite the overall joint density one more time as

$$\prod_{i=1}^{m_1} \left[\frac{\lambda_{t_i^1}}{\sum_{k \in P_i^1} \lambda_k} \cdot \text{Exp}\left(e_{t_i^1} \mid \sum_{k \in P_i^1} \lambda_k\right)\right] \cdot \mathbb{I}(t^{>1} = T^{>1}(e^2)) \cdot \prod_{k \in K} \text{Exp}\left(e_k^2 \mid \lambda_k^2\right), \tag{47}$$

where the last two terms have the same form as the joint density, written in the beginning, but for the execution trace variables $T^{>1}$ and transformed exponential variables $E^2$.

We use this observation and apply the same transformations to the density

$$\mathbb{I}(t^{>1} = T^{>1}(e^2)) \cdot \prod_{k \in K} \text{Exp}\left(e_k^2 \mid \lambda_k^2\right) \tag{48}$$

based on the partition $P_1^j, \ldots, P_{m_j}^j$ of $K^j \subset K$ for the next recursion steps. We apply the transformations until we reach the bottom of the recursion. By design, the algorithm excludes some of the indices from consideration $K^{j+1} \subsetneq K^j \subseteq K$. According to our notation, the variables excluded from the index set on a certain depth $j$ stay unchanged along with parameters, i.e. $E_k^j = E_k^{j'}$, $\lambda_k^j = \lambda_k^{j'}$ for $j' \geq j$. Such notation allows to preserve the product across all keys $\prod_{k \in K} \mathrm{Exp}(e_k^j \mid \lambda_k^j)$ throughout the recursion.

After performing all transformations at recursion depths $j$ from 2 to $N$ we arrive at the following representation of the joint density:

$$\prod_{j=1}^N \prod_{i=1}^{m_j} \left[ \frac{\lambda_{t_i^j}^j}{\sum_{k \in P_i^j} \lambda_k^j} \cdot \mathrm{Exp}\left( e_{t_i^j}^j \Big| \sum_{k \in P_i^j} \lambda_k^j \right) \right] \cdot \prod_{k \in K} \mathrm{Exp}\left( e_k^{N+1} \mid \lambda_k^{N+1} \right). \tag{49}$$

For each $k$ we observe one more time that $e_k^j$ and $\lambda_k^j$ do not change after $j = N(k) + 1$, since $k$ is excluded from all the corresponding $K^j$. Given this, we rewrite the latter product as

$$\prod_{k \in K} \mathrm{Exp}\left( e_k^{N(k)+1} \mid \lambda_k^{N(k)+1} \right). \tag{50}$$

Finally, we recursively apply the definition of $e_k^j$ to represent it as a function of the initial variable $e_k^1 = e_k$ and the set of minima $e_{t_{i(k,j')}^{j'}}^{j'}$, obtained at recursion depths $j'$ from 1 to $j - 1$. For each depth $j$:

$$e_k^{j+1} = e_k^j - e_{t_{i(k,j)}^j}^j = e_k^{j-1} - e_{t_{i(k,j)}^j}^j - e_{t_{i(k,j-1)}^{j-1}}^{j-1} = \ldots = e_k - \sum_{j'=1}^j e_{t_{i(k,j')}^{j'}}^{j'}. \tag{51}$$

Applying the same observation for $j = N(k)$, we obtain:

$$e_k^{N(k)+1} = e_k - \sum_{j=1}^{N(k)} e_{t_{i(k,j)}^j}^j, \tag{52}$$

which leads to the final representation of the joint density:

$$\underbrace{\left[ \prod_{j=1}^N \prod_{i=1}^{m_j} \frac{\lambda_{t_i^j}^j}{\sum_{k \in P_i^j} \lambda_k^j} \right]}_{\mathbb{P}_T(t;\lambda)} \cdot \underbrace{\left[ \prod_{j=1}^N \prod_{i=1}^{m_j} \mathrm{Exp}\left( e_{t_i^j}^j \Big| \sum_{k \in P_i^j} \lambda_k^j \right) \right]}_{\mathbb{P}_{E_T|T}(e_t|t;\lambda)} \cdot \underbrace{\prod_{k \in K} \mathrm{Exp}\left( e_k - \sum_{j=1}^{N(k)} e_{t_{i(k,j)}^j}^j \Big| \lambda_k^{N(k)+1} \right)}_{\mathbb{P}_{E|T,E_T}(e|t,e_t;\lambda)}. \tag{53}$$

This representation defines the following generation process:

- First, the trace variables (argminima) are generated from $\mathbb{P}_T(t; \lambda)$, the marginal probability of $T$, represented as the product of conditional probabilities of $T_i^j$;

- Second, the corresponding minima for all partition indices $i$ at all recursion depths $j$ are sampled from $\mathbb{P}_{E_T|T}(e_t \mid t; \lambda)$;

- Finally, the set of exponential random variables $E$ is obtained by sampling from $\mathbb{P}_{E|T,E_T}(e \mid t, e_t; \lambda)$. All the realizations $e_k$ here come from the exponential distribution with parameter $\lambda_k^{N(k)+1}$, shifted at the value $\sum_{j=1}^{N(k)} e_{t_{i(k,j)}^j}^j$.

Note that we have started from the joint distribution on $E, T$ and come to the joint distribution on $T, E_T, E$. We obtained larger set of variables, however, as initially, only $|K|$ of them are nondegenerate. This comes from the definition of $\lambda_k^j$ and the observation that at each step of taking

minimum we either find a constant zero, which does not change the number of non-constant variables, or find a non-degenerate value, introduce a new (non-degenerate) exponential variable, corresponding to the minimum, and replace the corresponding $\lambda_k^j$ with $+\infty$. The latter corresponds to setting one of the variables to be constant, thus, the overall number of non-degenerate distributions does not change when we perform the above transformations with density.

The first item above proves the claim about the distribution of trace variables. The second tells that each minimum realization $e_{t_{i(k,j)}^j}^j$ comes from the distribution $\mathrm{Exp}\left( \sum_{k' \in P_{i(k,j)}^j} \lambda_{k'}^j \right)$. Combined with the third one, it proves that the conditional distribution of each $E_k$ is the sum of the corresponding exponential distributions, claimed in the thorem:

$$E_k \mid T \sim \sum_{j=1}^{N(k)} \mathrm{Exp}\left( \sum_{k' \in P_{i(k,j)}^j} \lambda_{k'}^j \right) + \mathrm{Exp}\left( \lambda_k^{N(k)+1} \right). \tag{54}$$

$\square$

Based on the above derivation, we propose a procedure to compute the log-probability of the trace and to draw the conditional sample.

To compute the log-probability, we compute the log-probabilites of the top trace level $\{t_i^1\}_i$ as in Eq. 44. Then we repeat the exp-min trick as in the above derivation and repeat the procedure. Assume the induction hypothesis that the procedure computes the log-prob of the rest of the trace. Then, by induction, we obtain the log-prob of the whole trace as a sum of the log-prob of the top trace $\{t_i^1\}_i$ and the rest of the trace $\{t^{>1}\}_j$.

Similarly, assume we have a procedure to draw $e_k', k \in K'$. At the bottom of the recursion $e_k'$ are just exponential random variables. For the induction step, we draw $E_k$ for $k \notin K \setminus K'$ and definee $e_k := e_k' + e_{t_i}, k \in P_i$. In the next section, we provide the pseudo-code for the two procedures.

---

**Algorithm 4** $F_{\mathrm{struct}}(E, K, R)$ - returns structured variable $X$ based on utilities $E$ and auxiliary variables $K$ and $R$

---

**Input:** $E, K, R$
**Output:** $X$
  **if** $f_{\mathrm{stop}}(K, R)$ **then**
    **return**
  **end if**
  $P_1, \ldots, P_m \Leftarrow f_{\mathrm{split}}(K, R)$                     $\{\sqcup_{i=1}^m P_i = K\}$
  **for** $i = 1$ to $m$ **do**
    $T_i \Leftarrow \arg\min_{k \in P_i} E_k$
    **for** $k \in P_i$ **do**
      $E_k' \Leftarrow E_k - E_{T_i}$
    **end for**
  **end for**
  $K', R' \Leftarrow f_{\mathrm{map}}(K, R, \{T_i\}_{i=1}^m)$               $\{K' \subsetneq K\}$
  $E' \Leftarrow \{E_k' \mid k \in K'\}$
  $X' \Leftarrow F_{\mathrm{struct}}(E', K', R')$                    $\{\text{Recursive call}\}$
  **return** $f_{\mathrm{combine}}(X', K, R, \{T_i\}_{i=1}^m)$

---

---

**Algorithm 5** $F_{\text{log-prob}}(t, \lambda, K, R)$ - returns $\log \mathbb{P}_T(t; \lambda)$ for trace $t$, rates $\lambda$, $K$ and $R$ as in Alg. 2

---

**Input:** $t, \lambda, K, R$
**Output:** $\log \mathbb{P}_T(t; \lambda)$
  **if** $f_{\text{stop}}(K, R)$ **then**
    **return**
  **end if**
  $P_1, \ldots, P_m \Leftarrow f_{\text{split}}(K, R)$
  **for** $i = 1$ to $m$ **do**
    $\log \mathbb{P}_T(t_i^1; \lambda) \Leftarrow \log \lambda_{t_i^1} - \log \left( \sum_{k \in P_i} \lambda_k \right)$             {Index $j$ in $T_i^j$ denotes the recursion level}
    **for** $k \in P_i \setminus \{t_i^1\}$ **do**
      $\lambda_k' \Leftarrow \lambda_k$
    **end for**
    $\lambda_{t_i^1}' \Leftarrow +\infty$                                       {Because $E'(t_i^1) = 0$}
  **end for**
  $K', R' \Leftarrow f_{\text{map}}(K, R, \{t_i^1\}_{i=1}^m)$
  $\lambda' \Leftarrow \{\lambda_k' \mid k \in K'\}$
  $\log \mathbb{P}_T(t^{>1} \mid T^1 = t^1; \lambda) \Leftarrow F_{\text{log-prob}}(t^{>1}, \lambda', K', R')$     {Compute log-prob of $t^{>1} := \{t_i^j\}_{j>1}$}
  **return** $\sum_{i=1}^m \log \mathbb{P}_T(t_i^1; \lambda) + \log \mathbb{P}_T(t^{>1} \mid T^1 = t^1; \lambda)$

---

---

**Algorithm 6** $F_{\text{cond}}(t, \lambda, K, R)$ - returns a utility sample from $E \mid T = t, \lambda$ with rates $\lambda$ conditioned on the execution trace $t = \{t_i^j\}_{ij}$

---

**Input:** $t, \lambda, K, R$
**Output:** $e$
  **if** $f_{\text{stop}}(K, R)$ **then**
    **return**
  **end if**
  $P_1, \ldots, P_m \Leftarrow f_{\text{split}}(K, R)$
  **for** $i = 1$ to $m$ **do**
    $e_{t_i^1} \sim \text{Exp} \left( \sum_{k \in P_i} \lambda_k \right)$                                     {Sample the min}
    **for** $k \in P_i \setminus \{t_i^1\}$ **do**
      $\lambda_k' \Leftarrow \lambda_k$
    **end for**
    $\lambda_{t_i^1}' \Leftarrow +\infty$                                      {Because $e_{t_i^1}' = 0$}
  **end for**
  $K', R' \Leftarrow f_{\text{map}}(K, R, \{t_i^1\}_{i=1}^m)$
  $\lambda' \Leftarrow \{\lambda_k \mid k \in K'\}$
  $e' \Leftarrow F_{\text{cond}}(t^{>1}, \lambda', K', R')$       {Recursion, returns random variables indexed with $K'$}
  **for** $k \in K \setminus K'$ **do**
    $e_k' \sim \text{Exp}(\lambda_k')$                                 {Sample the rest of the utilities}
  **end for**
  **for** $i = 1$ to $m$ **do**
    **for** $k \in P_i \setminus \{t_i^1\}$ **do**
      $e_k \Leftarrow e_k' + e_{t_i^1}$                           {Reverse the Exponential-Min trick}
    **end for**
  **end for**
  **return** $e$

---

## B General Algorithms for Log-Probability and Conditional Sampling

We provide pseudo-code for computing $\log \mathbb{P}(T; \lambda)$ in Algorithm 3 and sampling $E \mid T$ in Algorithm 6. Both algorithms modify Algorithm 2 and use the same subroutines $f_{\text{stop}}$, $f_{\text{split}}$, $f_{\text{map}}$, and $f_{\text{combine}}$. Algorithms 3, 6 follow the structure as Algorithm 2 and have at most linear overhead in time and memory for processing variables such as $\lambda'$ and $\log \mathbb{P}(T_i^j \mid \lambda)$.

The indexed set of exponential random variables $E$ and the indexed set of the random variable parameters $\lambda$ have the same indices of indices $K$, which allows to call subroutines in the same way as in Algorithm 2.

Both algorithms take the trace variable $t = \{t_i^j\}_{j,i}$ as input. Note that index $j$ enumerate recursion levels. Both algorithms process the trace of the top recursion level $t_1^1, \ldots, t_m^1$ and make a recursive call to process the subsequent trace $t^{>1} := \{t_i^j\}_{i,j>1}$.

## C  Implementation Details

In the paper, we chose the exponential random variables and the recursive form of Algorithm 2 to simplify the notation. In practice, we parameterized the rate of the exponential distributions as $\lambda = \exp(-\theta)$, where $\theta$ was either a parameter or an output of a neural network. The parameter $\theta$ is essentially the location parameter of the Gumbel distribution and, unlike $\lambda$, can take any value in $\mathbb{R}$.

Additionally, the recursive form of Algorithm 2 does not facilitate parallel batch computation. In particular, the recursion depth and the decrease in size of $E$ may be different for different objects in the batch. Therefore, Algorithms 3,6 may require further optimization.

For the top-k algorithm, we implemented the parallel batch version. To keep the input size the same, we masked the omitted random variables with $+\infty$. We modeled the recursion using an auxiliary tensor dimension.

For the Kruskal's algorithm, we implemented the parallel batch version and used the $+\infty$ masks to preserve the set size. We rewrote the recursion as a Python for loop.

To avoid the computation overhead for the Chu-Liu-Edmonds algorithm, we implemented the algorithms in C++ and processed the batch items one-by-one.

For the binary trees Algorithm 2 was implemented in C++ and processed the batch items one-by-one, while Algorithms 3, 6 utilize efficient parallel implementation.

Also, during optimization using RELAX gradient estimator we observed the following behaviour: sometimes $E \mid T = t$ generates samples which do not lead to $t$ applying Algorithm 2. Such behaviour occurs due to the usage of *float* precision and does not show using *double* precision. While it may be considered as a drawback, its worth noting that it occurs very rare (less than 0.1 % of all conditional samples produced during optimization) and does not affect overall optimization procedure.

## D  Experimental Details

Setting up the experiments with Top-K, Spanning Tree and Arborescence we followed details about data generation, models and training procedures, described by [34], to make a valid comparison of the proposed score function methods with Stochastic Softmax Tricks (SSTs). In each experiment we fixed the number of function evaluations $N$ per iteration instead of batch size to make a more accurate comparison in terms of computational resources. With $N$ fixed, RELAX and SST were trained with batch size equal to $N$, while E-REINFORCE+ and T-REINFORCE+ were trained with batch size $N/K$ and $K$ samples of the latent structure for each object.

To get rid of the influence of any factors other than efficacy of the gradient estimator we fixed the same random model initialization. Then, for each gradient estimator we chose best model hyperparameter's set with respect to validation task metric (MSE, ELBO, accuracy). Given best model hyperparameter's set we report mean and standard deviations of the metrics across different random model initializations.

### D.1  Top-K and Beer Advocate

#### D.1.1  Data

As a base, we used the BeerAdvocate [30] dataset, which consists of beer reviews and ratings for different aspects: Aroma, Taste, Palate and Appearance. In particular, we took its decorrelated subset along with the pretrained embeddings from [27]. Every review was cut to $350$ embeddings, aspect ratings were normalized to $[0, 1]$.

Table 4: Results of $k$-subset selection on Appearance aspect data. MSE ($\times 10^{-2}$) and subset precision (%) is shown for best models selected on validation averaged across different random seeds.

| MODEL | ESTIMATOR | $k=5$ | | $k=10$ | | $k=15$ | |
|---|---|---|---|---|---|---|---|
| | | MEAN ± STD | PREC. | MEAN ± STD | PREC. | MEAN ± STD | PREC. |
| SIMPLE | *SST (Our Impl.)* | $3.44 \pm 0.13$ | $43.3 \pm 4.5$ | $3.09 \pm 0.12$ | $45.7 \pm 3.6$ | $\mathbf{2.67 \pm 0.12}$ | $\mathbf{42.1 \pm 1.1}$ |
| | *E-REINFORCE+* | $3.74 \pm 0.11$ | $38.8 \pm 2.9$ | $3.46 \pm 0.12$ | $33.2 \pm 3.6$ | $3.24 \pm 0.15$ | $31.2 \pm 3.4$ |
| | *T-REINFORCE+* | $3.57 \pm 0.11$ | $\mathbf{48.9 \pm 2.5}$ | $3.02 \pm 0.11$ | $\mathbf{47 \pm 4.1}$ | $2.69 \pm 0.06$ | $41.6 \pm 2.2$ |
| | *RELAX* | $\mathbf{3.36 \pm 0.1}$ | $44.2 \pm 3.2$ | $\mathbf{3.01 \pm 0.08}$ | $42.4 \pm 2.7$ | $2.85 \pm 0.09$ | $40.7 \pm 1.8$ |
| COMPLEX | *SST (Our Impl.)* | $2.96 \pm 1.1$ | $73.2 \pm 5.3$ | $2.61 \pm 0.09$ | $71.9 \pm 3.3$ | $2.57 \pm 0.08$ | $65.6 \pm 2.9$ |
| | *E-REINFORCE+* | $3.25 \pm 0.11$ | $72.9 \pm 6.1$ | $2.9 \pm 0.19$ | $63.1 \pm 1$ | $2.63 \pm 0.13$ | $63.3 \pm 0.5$ |
| | *T-REINFORCE+* | $\mathbf{2.65 \pm 0.05}$ | $\mathbf{82.9 \pm 1.3}$ | $\mathbf{2.48 \pm 0.05}$ | $74.5 \pm 3.7$ | $\mathbf{2.41 \pm 0.03}$ | $\mathbf{68.3 \pm 2}$ |
| | *RELAX* | $2.67 \pm 0.06$ | $81.3 \pm 1.5$ | $2.54 \pm 0.03$ | $\mathbf{74.8 \pm 1.3}$ | $2.51 \pm 0.03$ | $67.1 \pm 2.1$ |

Table 5: Results of $k$-subset selection on Taste aspect data. MSE ($\times 10^{-2}$) and subset precision (%) is shown for best models selected on validation averaged across different random seeds.

| MODEL | ESTIMATOR | $k=5$ | | $k=10$ | | $k=15$ | |
|---|---|---|---|---|---|---|---|
| | | MEAN ± STD | PREC. | MEAN ± STD | PREC. | MEAN ± STD | PREC. |
| SIMPLE | *SST (Our Impl.)* | $\mathbf{3.19 \pm 0.16}$ | $26.7 \pm 2.5$ | $\mathbf{2.93 \pm 0.12}$ | $28 \pm 0.9$ | $\mathbf{2.89 \pm 0.04}$ | $28.7 \pm 1.3$ |
| | *E-REINFORCE+* | $3.6 \pm 0.4$ | $23.6 \pm 2.6$ | $3.51 \pm 0.36$ | $21.4 \pm 2.2$ | $3.12 \pm 0.16$ | $24.6 \pm 3.2$ |
| | *T-REINFORCE+* | $3.24 \pm 0.2$ | $\mathbf{28.5 \pm 2.4}$ | $3.07 \pm 0.05$ | $\mathbf{28.5 \pm 1.4}$ | $2.9 \pm 0.04$ | $\mathbf{29.2 \pm 3.2}$ |
| | *RELAX* | $3.26 \pm 0.08$ | $24 \pm 3.4$ | $3.13 \pm 0.09$ | $25.8 \pm 2.1$ | $2.95 \pm 0.09$ | $24.4 \pm 2.6$ |
| COMPLEX | *SST (Our Impl.)* | $2.7 \pm 0.21$ | $36.2 \pm 3.1$ | $2.66 \pm 0.19$ | $36 \pm 5.1$ | $\mathbf{2.2 \pm 0.02}$ | $\mathbf{43.2 \pm 1}$ |
| | *E-REINFORCE+* | $3.43 \pm 0.52$ | $33.2 \pm 4.8$ | $3.15 \pm 0.33$ | $33 \pm 4.3$ | $2.81 \pm 0.16$ | $39.1 \pm 3$ |
| | *T-REINFORCE+* | $\mathbf{2.62 \pm 0.2}$ | $\mathbf{40.2 \pm 2.4}$ | $\mathbf{2.45 \pm 0.04}$ | $\mathbf{40.6 \pm 2.6}$ | $2.43 \pm 0.04$ | $40.3 \pm 2.3$ |
| | *RELAX* | $2.78 \pm 0.07$ | $34.7 \pm 2.5$ | $2.99 \pm 0.2$ | $32.1 \pm 3.6$ | $2.64 \pm 0.04$ | $33.9 \pm 3.8$ |

Table 6: Results of $k$-subset selection on Palate aspect data. MSE ($\times 10^{-2}$) and subset precision (%) is shown for best models selected on validation averaged across different random seeds.

| MODEL | ESTIMATOR | $k=5$ | | $k=10$ | | $k=15$ | |
|---|---|---|---|---|---|---|---|
| | | MEAN ± STD | PREC. | MEAN ± STD | PREC. | MEAN ± STD | PREC. |
| SIMPLE | *SST (Our Impl.)* | $\mathbf{3.63 \pm 0.17}$ | $\mathbf{28.1 \pm 2.7}$ | $3.37 \pm 0.08$ | $25 \pm 1.2$ | $3.14 \pm 0.09$ | $22.1 \pm 1.3$ |
| | *E-REINFORCE+* | $4.15 \pm 0.22$ | $21.3 \pm 6.3$ | $3.79 \pm 0.23$ | $19.6 \pm 3.1$ | $3.71 \pm 0.22$ | $15.8 \pm 2.1$ |
| | *T-REINFORCE+* | $3.81 \pm 0.2$ | $26.7 \pm 3.8$ | $\mathbf{3.33 \pm 0.09}$ | $\mathbf{26.9 \pm 1.1}$ | $\mathbf{3.14 \pm 0.07}$ | $21.6 \pm 1.2$ |
| | *RELAX* | $3.79 \pm 0.18$ | $26.8 \pm 3.4$ | $3.45 \pm 0.11$ | $23.6 \pm 1.6$ | $3.32 \pm 0.1$ | $\mathbf{22.3 \pm 1.2}$ |
| COMPLEX | *SST (Our Impl.)* | $2.98 \pm 0.09$ | $53.6 \pm 1$ | $\mathbf{2.79 \pm 0.01}$ | $45 \pm 1.2$ | $\mathbf{2.75 \pm 0.03}$ | $37.2 \pm 1.3$ |
| | *E-REINFORCE+* | $3.48 \pm 0.22$ | $47.3 \pm 5.3$ | $3.22 \pm 0.2$ | $39.7 \pm 3$ | $2.96 \pm 0.06$ | $36.8 \pm 3.2$ |
| | *T-REINFORCE+* | $\mathbf{2.92 \pm 0.03}$ | $\mathbf{56.3 \pm 0.8}$ | $2.87 \pm 0.03$ | $\mathbf{47.5 \pm 1.9}$ | $2.82 \pm 0.06$ | $\mathbf{40.4 \pm 1.8}$ |
| | *RELAX* | $3.05 \pm 0.03$ | $52.6 \pm 1.9$ | $3.03 \pm 0.09$ | $42.6 \pm 3.6$ | $2.86 \pm 0.05$ | $36.6 \pm 1.2$ |

### D.1.2 Model

We used the Simple and Complex models defined by [34] for parameterizing the mask. The Simple model architecture consisted of Dropout (with $p = 0.1$) and a one-layered convolution with one kernel. In the Complex model architecture, two more convolutional layers with 100 filters and kernels of size 3 were added.

### D.1.3 Training

We trained all models for 10 epochs with $N = 100$. We used the same hyperparameters ranges as in [34], where it was possible. Hyperparameters for our training procedure were learning rate, final decay factor, weight decay. They were sampled from $\{1, 3, 5, 10, 30, 50, 100\} \times 10^{-4}$, $\{1, 10, 100, 1000\} \times 10^{-4}$, $\{0, 1, 10, 100\} \times 10^{-6}$ respectively. We also considered regularizer type for SST as hyperparameter ($\{$Euclid., Cat. Ent., Bin. Ent., E.F. Ent.$\}$). For $E$-REINFORCE+ and $T$-REINFORCE+

number of latent samples for every example in a batch was considered as hyperparameter with range $\{1, 2, 4\}$. We tuned hyperapameters over considered ranges with uniform search with 25 trials. Best model were chosen with respect to best validation MSE.

Results for Appearance aspect can be found in Table 4, for Taste aspect in Table 5, for Palate aspect in Table 6. Mean and standard deviations reported in the tables are computed across 16 different random model initializations. In conclusion, we can state that the proposed method is comparable with SST on BeerAdvocate dataset.

## D.2 Spanning Tree and Graph Layout

Table 7: Graph Layout experiment results for T=20 iterations. Metrics are obtained by choosing models with best validation ELBO and averaging results across different random seeds on the test set.

| | $T = 20$ | | | | | |
|---|---|---|---|---|---|---|
| ESTIMATOR | ELBO | | EDGE PREC. | | EDGE REC. | |
| | MEAN $\pm$ STD | MAX | MEAN $\pm$ STD | MAX | MEAN $\pm$ STD | MAX |
| *SST (Our Impl.)* | $-2039.42 \pm 1079.56$ | $-1483.31$ | $83 \pm 30$ | 98 | $93 \pm 9$ | 98 |
| *T-REINFORCE+* | $-1976.16 \pm 980.12$ | $-1458.81$ | $83 \pm 30$ | 98 | $94 \pm 8$ | 98 |
| *RELAX* | $-3129.51 \pm 1464.88$ | $-1594.85$ | $60 \pm 37$ | 98 | $90 \pm 8$ | 98 |

### D.2.1 Data

For each dataset entry we obtained the corresponding ground truth spanning tree by sampling a fully-connected graph on 10 nodes and applying Kruskal algorithm. Graph weights were sampled independently from Gumbel$(0, 1)$ distribution. Initial vertex locations in $\mathbb{R}^2$ were distributed according to $N(0, I)$. Given the spanning tree and initial vertex locations, we applied the force-directed algorithm [8] for $T = 10$ or $T = 20$ iterations to obtain system dynamics. We dropped starting positions and represented each dataset entry as the obtained sequence of $T = 10$ or $T = 20$ observations. We generated 50000 examples for the training set and 10000 examples for the validation and test sets.

### D.2.2 Model

Following [34] we used the NRI model with encoder and decoder architectures analogous to the MLP encoder and MLP decoder defined by [17].

**Encoder.** Given the observation of dynamics, GNN encoder passed messages over the fully connected graph. Denoting its final edge representation by $\theta$, we obtained parameters of the distribution over undirected graphs as $\frac{1}{2}(\theta_{ij} + \theta_{ji})$ for an edge $i \leftrightarrow j$. Hard samples of spanning trees were then obtained by applying the Kruskal algorithm on the perturbed symmetrized matrix of parameters $\lambda_{ij} = \exp\left(-\frac{1}{2}(\theta_{ij} + \theta_{ji})\right)$.

**Decoder.** GNN decoder took observations from previous timesteps and the adjacency matrix $X$ of the obtained spanning tree as its input. It passed messages over the latent tree aiming at predicting future locations of the vertices. We used two separate networks to send messages over two different connection types ($X_{ij} = 0$ and $X_{ij} = 1$). Since parameterization of the model was ambiguous in terms of choosing the correct graph between $X$ and $1 - X$, we measured structure metrics with respect to both representations and reported them for the graph with higher edge precision.

In experiments with RELAX we needed to define a critic. It was a simple neural network defined as an MLP which took observations concatenated with the perturbed weights and output a scalar. It had one hidden layer and ReLU activations.

**Objective.** During training we maximized ELBO (lower bound on the observations' log-probability) with gaussian log-likelihood and KL divergence measured in the continuous space of exponential noise. It resulted in an objective which was also a lower bound on ELBO with KL divergence measured with respect to the discrete distributions.

### D.2.3  Training

We fixed the number of function evaluations per iteration at $N = 128$. All models were trained for 50000 iterations. We used constant learning rates and Adam optimizer with default hyperparameters. For all estimators we tuned separate learning rates for encoder, decoder and RELAX critic by uniform sampling from the range $[1, 100] \times 10^{-5}$ in log scale. Additionally, for $T$-REINFORCE+ we tuned $K$ in $\{2, 4, 8, 16\}$ and for RELAX we tuned size of the critic hidden layer in $\{256, 512, 1024, 2056\}$. We did not train $E$-REINFORCE+ since [34, Appendix, Sec. C.1] report its bad performance (REINFORCE (*Multi-sample*) according to their namings). We used Gumbel Spanning Tree SST because it showed the best performance on the corresponding task in [34, Section 8.1]. We tuned hyperapameters over considered ranges with uniform search with 20 trials. Best model were chosen with respect to best validation ELBO.

Mean and standard deviations reported in the tables are computed across 10 different random model initializations. Table 7 reports results for T=20 iterations. Despite the fact that the dataset for this experiment is highly synthetic we can note that model initialization plays big role in the final performance of the gradient estimator. Overall, we can see that $T$-REINFORCE+ performs slightly better in terms of ELBO which is expected since score function based methods give unbiased gradients of ELBO, while relaxation-based SST optimizes relaxed objective.

### D.3  Arborescence and Unsupervised Parsing

#### D.3.1  Data

We took the ListOps [32] dataset, containing arithmetical prefix expressions, e.g. `min[3 med[3 5 4] 2]`, as a base, and modified its sampling procedure. We considered only the examples of length in $[10, 50]$ that do not include the `summod` operator and have bounded depth $d$. Depth was measured with respect to the ground truth parse tree, defined as a directed graph with edges going from functions to their arguments. We generated equal number of examples for each $d$ in $\{1, \ldots, 5\}$. Train dataset contained 100000 samples, validation and test sets contained 20000 samples.

#### D.3.2  Model

Model mainly consisted of two parts which we call encoder and classifier.

Encoder was the pair of identical left-to-right LSTMs with one layer, hidden size 60 and dropout probability 0.1. Both LSTMs used the same embedding lookup table. Matrices that they produced by encoding the whole sequence were multiplied to get parameters of the distribution over latent graphs. Equivalently, parameter for the weight of the edge $i \to j$ was computed as $\theta_{ij} = \langle v_i, w_j \rangle$, where $v_i$ and $w_j$ are hidden vectors of the corresponding LSTMs at timesteps $i$ and $j$. Given $\lambda = \exp(-\theta) \in \mathbb{R}^{n \times n}$, we sampled matrix weights from the corresponding factorized exponential distribution. Hard samples of latent arborescences, rooted at the first token, were obtained by applying Chu-Liu Edmonds algorithm to the weighted graph.

Classifier mainly consisted of the graph neural network which had the initial sequence embedding as an input and ran 5 message sending iterations over the sampled arborescence's adjacency matrix. It had its own embedding layer different from used in the encoder. GNN's architecture was based on the MLP decoder model by [17]. It had a two-layered MLP and did not include the last MLP after message passing steps. Output of the GNN was the final embedding of the first token which was passed to the last MLP with one hidden layer. All MLPs included ReLU activations and dropout with probability 0.1.

In experiments with RELAX we needed to define a critic. It contained LSTM used for encoding of the initial sequence. It was left-to-right, had a single layer with hidden size 60 and dropout probability 0.1. It had its own embedding lookup table. LSTM's output corresponding to the last token of the input sequence was concatenated with a sample of the graph adjacency matrix and fed into the output MLP with one hidden layer of size 60 and ReLU activations. Before being passed to the MLP, weights of the adjacency matrix were centered and normalized.

### D.3.3 Training

We fixed the number of function evaluations per iteration at $N = 100$ and trained models for 50000 iterations. We used AdamW optimizer, separate for each part of the model: encoder, classifier and critic in case of RELAX. They all had constant, but not equal in general case, learning rates, and default hyperparameters. We used Gumbel arborescence SST because it showed the best performance on the corresponding task in [34, Section 8.2]. We tuned learning rates and weight decays in range $[1, 100] \times 10^{-5}$ in log scale and the number of latent samples in $\{2, 4, 5\}$ for $E$-REINFORCE+ and $T$-REINFORCE+. We tuned hyperapameters over considered ranges with uniform search with 20 trials. Best model were chosen with respect to best validation accuracy.

Table 3 with results indicates more stable performance of score function based gradient estimators with respect to different random model initializations.

### D.4 Binary Tree and Non-monotonic Generation

#### D.4.1 Data

In this experiment, we constructed 5 datasets of balanced parentheses, varying the number of their types in $\{10, 20, \ldots, 50\}$. For each number of parentheses' types we constructed a dataset by generating independent sequences with the following procedure:

1. Sample length $l$ of the sequence from the uniform distribution on $\{2, 4, \ldots, 20\}$.
2. Uniformly choose current type of parentheses.
3. Choose one of the configurations `"( sub )"` or `"() sub"` with equal probabilities, where `"("` and `")"` denote the pair corresponding to the current parentheses type.
4. Make a recursive call to generate substring `sub` with length $l - 2$.
5. Return the obtained sequence.

Each train dataset contained 20000 samples, validation and test sets contained 2500 samples. In case of semi-supervised experiments, datasets were modified to contain 10% of supervision.

#### D.4.2 Model

Language model consisted of the decoder with non-monotonic architecture, defined in [44], and of the encoder (in case of semi-supervised training). In this experiment all models shared the same hidden and embedding dimensions equal to 300.

**Decoder.** We fixed decoder's architecture to be a single-layer left-to-right LSTM. While training, we processed a tree-ordered input by first adding leaf nodes, labeled with `EOS` token, to all places with a child missing, and transforming the modified tree into a sequence by applying the level-order traversal. The obtained sequence was then used for training in the teacher-forcing mode. While generating, we sampled raw sequences (treated as level-order traversals), transformed them into binary trees and output the in-order traversal of the obtained tree.

**Encoder.** For semi-supervised training we defined the encoder as a single-layer bidirectional LSTM. Given an input sequence of length $l$, it output a vector of exponential parameters $\lambda = (\lambda_1, \ldots, \lambda_l)$. Hard samples of latent trees were obtained by applying Algorithm 9 on the perturbed $\lambda$.

**Critic.** Critic, used for estimating encoder's gradients with RELAX, was defined as a single-layer bidirectional LSTM. It took a sequence, concatenated with perturbed output of the encoder along the embedding dimension, as its input, and output a single value.

#### D.4.3 Smart order

We defined *smart* order for binary trees in the way, visualized in Figure 3. Opening parentheses do not have left children, while their right children are fixed to be the corresponding closing parentheses. Construction starts from the first token; each time we generate a pair of parentheses and have a substring between them, we make a recursion step, generating the corresponding subtree at the left from the current closing parenthesis. If there is a substring at the right of the generated pair, corresponding subtree is attached as the closing parenthesis' right child.

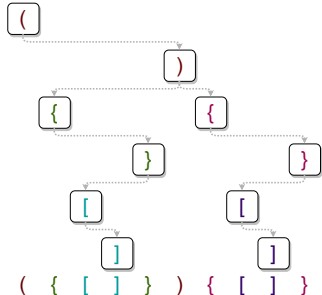

Figure 3: Visualization of *smart* order for binary trees

From decoder's perspective (level-order generation) this order corresponds to an altering process of generation, where blocks of opening parentheses are followed by the corresponding closing ones. Intuitively, this type of process should simplify producing balanced parentheses sequences, since we do not mix opening and closing parentheses at each stage.

### D.4.4 Training

Models with fixed order (*left-to-right* and *smart*) were trained by minimization of cross-entropy using teacher-forcing. Semi-supervised models were trained in a manner of variational autoencoders. Unsupervised part of the training objective was defined by ELBO, lower bound on the marginal likelihood of training sequences, while supervised part consisted of joint likelihood (of sequence and fixed order), defined as negative cross-entropy between the decoder's output and train sequences, and the encoder's likelihood of the *smart* order.

All models were trained for 50 epochs. We chose the best model by measuring perplexity of the validation set. It was calculated explicitly for fixed-order models and approximated by IWAE bound [1] for semi-supervised ones. We observed that distribution of the encoder became degenerate during optimization, while decoder did not follow this behaviour. It made IWAE estimation with variational distribution highly underestimated. Instead of variational distribution, we used the empirical distribution on orders, obtained by sampling 10000 trees from decoder. Number of latent samples for IWAE estimation was fixed at $K = 1000$.

Results from Table 2 suggest that generative metrics of the model can be improved by training on the non-trivial order of generation even using semi-supervised approach with relatively small amount of supervision.

### D.5 Permutations by argsort and Non-monotonic Generation

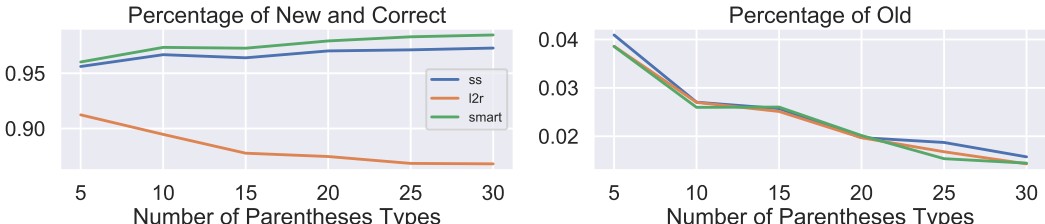

Figure 4: Generative statistics of insertion-based non-monotonic language model with different orders. Semi-supervised order is defined by Plackett-Luce distribution supervised with *smart* order.

To explore the applicability of the stochastic invariants to other models for non-monotonic text generation we consider the same task as in the previous experiment but examine another generative model as well as the latent variable which defines orderings.

### D.5.1 Data

We used the same data generative process as in the previous experiment (Appendix D.4.1).

### D.5.2 Model

Model utilizes encoder-decoder architecture. For the decoder we took Transformer-InDIGO model from [13]. It generates sequences by insertions using relative position representations. [28] discovered the one-to-one correspondence between relative positions and permutations, therefore we used them interchangeably in the model. Encoder is simple 1 layer bidirectional LSTM network which outputs parameters of the Plackett-Luce distribution given sequence of tokens. We used RELAX gradient estimator to train encoder parameters. Critic was also 1 layer bidirectional LSTM network which outputs scalar given concatenation of sequence of tokens and exponential noise.

### D.5.3 Training

We trained the model with different orders: *left-to-right*, *smart* and semi-supervised with 10% supervision with *smart* orders for 100 epochs. *Smart* order was defined by sequential generation of opening parenthesis and the corresponding closing parenthesis. Intuitively, it should be easier to generate balanced parentheses sequences using this order since model does not need stack to remember number of opened parentheses. Semi-supervised model was trained in the manner of semi-supervised variational autoencoders with teacher forcing for reconstruction term.

For each order we chose the best model with respect to the decoder perplexity (for semi-supervised we estimated marginal likelihood using IWAE estimator [1] with variational distribution as the proposal). During training we observed that different orders achieve the same perplexity which is expected since the data is too simple to model with any order. From the Figure 4 we can observe the same behaviour as with binary trees. While different orders achieve the same perplexity on the test set, using non-nomonotonic orders improves generation quality.

### D.6 Additional Tables

Table 8: Standard deviation (std) of the gradient estimators on Graph Layout experiment for T=10 iterations. Results are obtained by choosing models with best validation ELBO and averaging std estimates across *train* set. Standard deviation is estimated with 10 samples for each batch.

| | MEAN GRADIENT STD | | |
|---|---|---|---|
| ESTIMATOR | BEGINNING | 25K | 50K |
| *SST (Our Impl.)* | 0.1674 | 0.0343 | 0.0379 |
| *T-REINFORCE+* | 1.6320 | 1.4409 | 0.9924 |
| *RELAX* | 3.0592 | 1.0292 | 0.8874 |

Table 9: Time per one gradient update (ms) for different gradient estimators and structured variables.

| | TIME PER ITER (MS) | | |
|---|---|---|---|
| STRUCTURE | SST | T-REINFORCE+ | RELAX |
| *Spanning Tree* | 123 | 137 | 180 |
| *Arborescence* | 175 | 249 | 535 |

Table 10: Results of $k$-subset selection on Aroma aspect *train* data. MSE ($\times 10^{-2}$) is shown for best models selected on validation averaged across different random seeds.

| MODEL | ESTIMATOR | $k = 5$ MEAN $\pm$ STD | $k = 10$ MEAN $\pm$ STD | $k = 15$ MEAN $\pm$ STD |
|---|---|---|---|---|
| SIMPLE | *SST (Our Impl.)* | **3.22 $\pm$ 0.17** | 2.96 $\pm$ 0.14 | 2.54 $\pm$ 0.13 |
| | *E-REINFORCE+* | 3.45 $\pm$ 0.19 | 3.38 $\pm$ 0.23 | 2.85 $\pm$ 0.21 |
| | *T-REINFORCE+* | 3.38 $\pm$ 0.13 | **2.93 $\pm$ 0.19** | **2.43 $\pm$ 0.12** |
| | *RELAX* | 3.36 $\pm$ 0.12 | 3.13 $\pm$ 0.18 | 2.78 $\pm$ 0.17 |
| COMPLEX | *SST (Our Impl.)* | 2.69 $\pm$ 0.14 | 2.24 $\pm$ 0.18 | 2.18 $\pm$ 0.11 |
| | *E-REINFORCE+* | 2.85 $\pm$ 0.22 | 2.56 $\pm$ 0.25 | 2.44 $\pm$ 0.19 |
| | *T-REINFORCE+* | **2.5 $\pm$ 0.16** | **2.19 $\pm$ 0.15** | **2.09 $\pm$ 0.1** |
| | *RELAX* | 2.53 $\pm$ 0.15 | 2.21 $\pm$ 0.17 | 2.19 $\pm$ 0.14 |

Table 11: Graph Layout experiment results for T=10 iterations. Metrics are obtained by choosing models with best validation ELBO and averaging results across different random seeds on the *train* set.

| ESTIMATOR | $T = 10$ ELBO MEAN $\pm$ STD | MAX | EDGE PREC. MEAN $\pm$ STD | MAX | EDGE REC. MEAN $\pm$ STD | MAX |
|---|---|---|---|---|---|---|
| *SST (Our Impl.)* | $-1846.93 \pm 1124.23$ | $-1357.03$ | **88 $\pm$ 22** | **96** | **94 $\pm$ 4** | **96** |
| *T-REINFORCE+* | **$-1584.11 \pm 572.16$** | **$-1193.11$** | 70 $\pm$ 31 | 91 | 86 $\pm$ 8 | 91 |
| *RELAX* | $-2086.93 \pm 573.72$ | $-1207.83$ | 43 $\pm$ 31 | 90 | 81 $\pm$ 6 | 90 |

Table 12: Unsupervised Parsing on ListOps. We report the average *train*-performance of the model with the best validation accuracy across different random initializations.

| ESTIMATOR | ACCURACY MEAN $\pm$ STD | MAX | PRECISION MEAN $\pm$ STD | MAX | RECALL MEAN $\pm$ STD | MAX |
|---|---|---|---|---|---|---|
| *SST (Our Impl.)* | 79.31 $\pm$ 8.17 | **94.73** | 57.15 $\pm$ 19.92 | **82.51** | 30.58 $\pm$ 19.03 | 73.28 |
| *E-REINFORCE+* | 60.64 $\pm$ 2.51 | 65.21 | 41.12 $\pm$ 6.68 | 45.95 | 40.99 $\pm$ 6.75 | 45.68 |
| *T-REINFORCE+* | **88.69 $\pm$ 3.02** | 93.06 | **78.6 $\pm$ 7.37** | 80.68 | **61.78 $\pm$ 14.52** | **80.68** |
| *RELAX* | 79.92 $\pm$ 9.35 | 88.64 | 54.84 $\pm$ 17.51 | 75.27 | 53.73 $\pm$ 17.18 | 75.27 |

# E   Pseudo-Code

This section containes pseudo-code for the algorithms with stochastic invariants discussed in the paper.

## E.1   Pseudo-Code for $\arg \operatorname{top} k$

We refer the reader to Algorithm 1 in the main paper.

## E.2   Pseudo-Code for $\operatorname{argsort}$

Algorithm 7 presents a recursive algorithm for sorting. The algorithm implements the insertion sorting. Although insertion sorting may not be the most efficient sorting algorithm, it has a stochastic invariant. Indeed, the algorithm recursively finds the minimum element and then excludes the element from the consideration. As opposed to $\arg \operatorname{top} k$, the algorithm does not omit the order of $X'$. As a result, for the algorithm, the trace $T$ coincides with the output $X$.

**Algorithm 7** $F_{\text{sort}}(E, K)$ - sorts the set $K$ based on the corresponding $E$ values

---

**Input:** $E, K$
**Output:** $X$
  **if** $E = \emptyset$ **then**
    **return**
  **end if**
  $T \Leftarrow \arg\min_{j \in K} E_j$                                                {Find the smallest element}
  **for** $j \in K$ **do**
    $E'_j \Leftarrow E_j - E_T$
  **end for**
  $K' \Leftarrow K \setminus \{T\}$                                             {Exclude $\arg\min$ index $T$}
  $E' \Leftarrow \{E'_k \mid k \in K'\}$
  $X' \Leftarrow F_{\text{sort}}(E', K')$                                        {Sort the subset $K'$}
  **return** $(T, X'_1, \ldots, X'_{\text{size}(X')})$      {Concatenate $T$ and the subset sorting $X'$}

---

### E.3 Pseudo-Code for the Bespoke Matching Variable

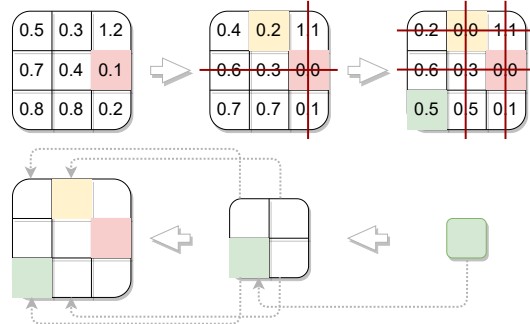

Figure 5: The generative process for perfect matchings. On the top row, the algorithm recursively finds the minimum element and excludes the corresponding row and column. On the bottom row, the algorithm iteratively combines the subset matching $X'$ and the minimum element $T$.

---

**Algorithm 8** $F_{\text{match}}(E, K)$ - returns a matching between the columns and the rows of a square matrix $E$ with the elements indexed with $K$.

---

**Input:** $E, K$
**Output:** $X$
  **if** $E = \emptyset$ **then**
    **return**
  **end if**
  $T \Leftarrow \arg\min_{(u,v) \in K} E_{(u,v)}$              {Find the smallest element, $T$ is an integer tuple}
  **for** $(u, v) \in K$ **do**
    $E'_{(u,v)} \Leftarrow E_{(u,v)} - E_T$
  **end for**
  {Cross out the row and the column containing the minimum}
  $K' \Leftarrow \{(u, v) \in K \mid u \neq T[0] \vee v \neq T[1]\}$
  $E' \Leftarrow \{E'_{(u,v)} \mid (u, v) \in K'\}$
  $X' \Leftarrow F_{\text{match}}(E', K')$                       {Find a matching for the sub-matrix}
  **return** $\{T\} \cup X'$                                 {Add an edge (tuple) to the matching}

---

We present an algorithm with the stochastic invariant that returns a matching between the rows and the columns of a square matrix. Although such a variable is in one-to-one correspondence with permutations, the distribution has more parameters than the Plackett-Luce distribution. We speculate that the distribution may be more suitable for representing finite one-to-one mappings with a latent

variable. Figure 5 illustrates the idea behind the algorithm. In particular, the algorithm iteratively finds the minimum element and excludes the row and the column containing the element from the matrix. Then the algorithm uses recursion to construct a matching for the submatrix.

Notably, we were unable to represent the Hungarian algorithm for the minimum matching to the problem. Algorithm 8 returns the same output when the column-wise minimum elements form a matching in the matrix. In general, the output matching may not be the minimum matching.

### E.4    Pseudo-Code for the Bespoke Binary Tree Variable

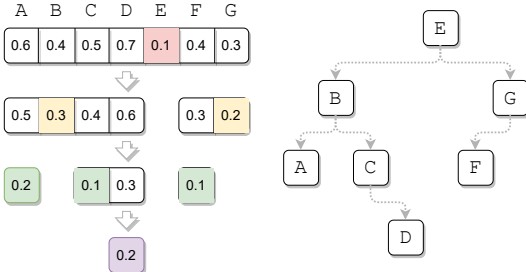

Figure 6: The generative process for binary trees. On the left, we assign weights to tokens and set the minimum weight token to be the root. Then we recursively construct trees for the tokens on the left-hand side and the right-hand side of the root. On the right, we present the resulting tree.

---

**Algorithm 9** $F_{\text{tree}}(E, K, R)$ - constructs a binary tree based on weight $E$ with the node set $K$. The auxiliary variable $R$ is a partition of nodes initialized as a single set $K$.

---

**Input:** $E, K, R$
**Output:** $X$
  **if** $E = \emptyset$ **then**
    **return**
  **end if**
  $P_1, \ldots, P_m \Leftarrow R$
  **for** $i = 1$ to $m$ **do**
    $T_i \Leftarrow \arg\min_{j \in P_i} E_j$
    **for** $j \in P_i$ **do**
      $E'_j \Leftarrow E_j - E_{T_i}$
    **end for**
  **end for**
  {$R'$ splits the partition sets $P_i$ into the left-hand side and the right-hand side nodes relative to $T_i$}
  {for example, when $T = 5$ we split $P = \{3, 4, 5, 6, 7, 8\}$ into $\{3, 4\}$ and $\{6, 7, 8\}$}
  $R' \Leftarrow \{P_i[0, T_i) \mid i = 1, \ldots, m\} \cup \{P_i(T_i, -1] \mid i = 1, \ldots, m\}$
  $K' \Leftarrow K \setminus \{T_1, \ldots, T_m\}$
  $E' \Leftarrow \{E'_k \mid k \in K'\}$
  $X' \Leftarrow F_{\text{tree}}(E', K', R')$                    {Recursive call returns a sequence of $2m$ subtrees}
  **return** $\big((T_i, X'[2i], X'[2i + i]), i = 1, \ldots, m\big)$ {Join the $2m$ trees into $m$ with roots $T_1, \ldots, T_m$}

---

For our experiments with the non-monotonic generation, we propose a distribution over binary trees. Given a sequence of tokens, we assign an exponential random variable to each token. Then we construct a tree with the following procedure illustrated in Figure 6. First, we set the token with the minimum weight to be the root of the tree. The tokens on the left-hand side from the root will be the left descendants of the root, the tokens on the right-hand side will be the right descendants of the root. Then we repeat the procedure for the left-hand descendants and the right-hand descendants independently. We summarise the above in Algorithm 9.

Intuitively, the algorithm should include two recursive calls: one for the left-hand side subtree and the other for the right-hand side subtree. According to the general framework (see Algorithm 2), our pseudo-code is limited to a single recursive call. In particular, at each recursion depth $k$ Algorithm 9

processes all subtrees with a root at the given depth $k$. As a result, at depth $k$ the partition includes $m = 2^k$ subsets, some of which may be empty. Alternatively, Algorithm 2 can be extended to multiple recursive calls.

## E.5 Kruskal's Algorithm

Kruskal's algorithm [24, 5] for the minimum spanning tree gives another illustration of the framework. In this case, the edge weights are the exponential random variables. The input variable $E$ is indexed by the edges in a graph, i.e. $E_{(u,v)}$ is the weight of the edge $(u,v)$.

Algorithm 10 contains the pseudo-code for the Kruskal's algorithm. The auxiliary variable $R$ is a set of disjoint sets of nodes. It represents the connected components of the current subtree. The algorithm build the tree edge-by-edge. It starts with an empty set of edges and all sets in $R$ of size one. Then the algorithm finds the lightest edge connecting between the connected components of $R$ and joins the two connected components. The algorithm repeats this greedy strategy until $R$ contains a single connected component.

From the recursion viewpoint, the algorithm constructs a tree with the nodes being the elements of $R$. First, the algorithm adds the lightest edge $T = (u,v)$ to the tree. Then it joins the sets $R_u$ and $R_v$ containing $u$ and $v$, we denote the result as $R'$. Next, the algorithm uses the recursion to construct a tree where the nodes are the elements of $R'$. The size of $R$ decreases with each step, therefore the recursion will stop. The resulting tree is a tree for the connected components $R'$ along with the edge $T$.

Notably, Prim's algorithm is a similar greedy algorithm for finding the minimum spanning tree. However, the algorithm considers different subsets of edges; as a result, we could not represent Prim's algorithm as an instance of Algorithm 2.

---

**Algorithm 10** $F_{\text{Kruskal}}(E, K, R)$ - finds the minimum spanning tree given edges $K$ with the corresponding weights $E$; Call with $R := \{\{v\} \mid v \in V\}$ set as node singletons

---

**Input:** $E, K, R$
**Output:** $X$
  **if** $|R| = 1$ **then**
    **return**
  **end if**
  $T \Leftarrow \arg\min_{k \in K} E_k$                                          {Find the smallest edge}
  **for** $k \in K$ **do**
    $E'_k \Leftarrow E_k - E_T$
  **end for**
  {For $T = (u,v)$ find $R_u, R_v \in R$ s.t. $u \in R_u, v \in R_v$}
  $R_u, R_v \Leftarrow \text{find\_connected\_components}(R, T)$ {Merge the connected components $R_u$ and $R_v$}
  $R' \Leftarrow (R \setminus \{R_u, R_v\}) \cup (\{R_u \cup R_v\})$         {Remove edges connecting $R_u$ and $R_v$}
  $K' \Leftarrow K \setminus \{(u', v') \in K \mid u', v' \in R_u \cup R_v\}$
  $E' \Leftarrow \{E'_k \mid k \in K'\}$
  $X' \Leftarrow F_{\text{Kruskal}}(E', K', R')$          {Edges in $X'$ form a spanning tree for nodes in $R'$}
                                          {$X \cup \{T = (u,v)\}$ is a spanning tree for nodes in $R$}
  **return** $X' \cup \{T\}$

---

## E.6 Chu-Liu-Edmonds Algorithm

We adopt Chu-Liu-Edmonds algorithm from [18] in Algorithm 11. Similarly to Kruskal's algorithm, the perturbed input $E$ represents the weights of the graph edges, its indices are the edges of the input directed graph.

As opposed to the previous examples, the algorithm considers multiple subsets of indices $P_1, \ldots, P_m$ at each recursion level. In particular, for each node except $r$ the algorithm finds the incoming edge with minimal weight. If $\{T_i\}_{i \neq r}$ is an arborescence, $f_{\text{combine}}$ returns it.

Otherwise, $\{T_i\}_{i \neq r}$ contains a cycle and $f_{\text{map}}$ constructs a new graph with the cycle nodes contracted to a single node. Similarly to Kruskal's algorithm, we use the variable $R$ to store sets of nodes.

In this case, $R$ represent the contracted node as a set of the original nodes. To construct $X$, the subroutine $f_{\text{combine}}$ expands the contracted loop in $X'$ and adds all edges in $C$ but one.

---

**Algorithm 11** $F_{\text{CLE}}(E, K, R, r)$ - finds the minimum arborescence $X$ of a directed graph with edges $K$ of weight $E$ and root node $r$. Auxiliary variable $R$ is a partition of nodes indicating merged nodes, initialized with node singletons $R := \{\{v\} \mid v \in V\}$.

---

**Input:** $E, K, R, r$
**Output:** $X$
  **if** $|R| = 1$ **then**
    **return**
  **end if**
  $P_1, \ldots, P_m \Leftarrow f_{\text{split}}(K, R, r)$            {Split $K$ into sets of edges ending at $R_v \in R, r \notin R_v$;}
  **for** $i = 1$ to $m$ **do**
    $T_i \Leftarrow \arg\min_{k \in P_i} E_k$
    **for** $k \in P_i$ **do**
      $E'_k \Leftarrow E_k - E_{T_i}$
    **end for**
  **end for**
  $C \Leftarrow \text{find\_loop}(R, \{T_i\}_{i=1}^m)$           {Find a loop $C$ assuming nodes $R$ and edges $\{T_i\}_{i=1}^m$}
  **if** $C = \emptyset$ **then**
    **return** $\{T_i\}_{i=1}^m$
  **end if**
  $R' \Leftarrow (R \setminus \{C_i \mid C_i \in C\}) \cup (\cup_{i=1}^{|C|} C_i)$        {Contract the loop nodes into a single node}
  $K' \Leftarrow K \setminus \{(u, v) \in K \mid u \in C_i, v \in C_j\}$
  $E' \Leftarrow \{E'_k \mid k \in K'\}$
  $X' \Leftarrow F_{\text{CLE}}(E', K', R', r)$           {Find arborescence for the contracted graph}
  $X \Leftarrow X' \cup \{T_i \mid T_i \text{ in cycle, preserves arborescence}\}$      {Add to $X'$ all loop edges but one}
  **return** $X$

---