# OpenReview forum: "Leveraging Recursive Gumbel-Max Trick for Approximate Inference in Combinatorial Spaces"
_NeurIPS.cc/2021/Conference — NeurIPS 2021 Poster_

### Official Review · Reviewer_ahTk · 2021-07-11

**Rating:** 6
**Confidence:** 3

**Summary:**

The paper proposed a new gradient estimator, which is composed of Gumbel max top-k trick and the score function, for training discrete distribution.

**Limitations And Societal Impact:**

Yes, the limitations are addressed in section 3.3 and 6.

**Main Review:**

Originality: The gradient estimator is new.

Clarity: The paper is hard to follow.
1. I didn't find what's the meaning of "SST", which is treated as the baseline in all the experiments.
2. I couldn't understand what is "stochastic invariant", is the definition of "stochastic invariant" first appears in line 131: "the first argument E of the algorithm always has
132 exponential distribution (conditioned on the trace T from the above recursion levels)"?
3.  I think it would be better to move the introduction of  "Execution Trace T" to the beginning of section 3, otherwise, we cannot understand table 1.


Quality:
1. The proposed method needs a lot of computation, however, it has larger variance than the standard X space score function estimator. (line 205)  It is mentioned that the standard score function estimator is preferable when we can access the score function, otherwise, the proposed estimator is available. Could the author give a few practical examples where the score function is not available?


Significance:
I don't know the significance since I don't even know what the baseline method SSL represents for.



**Time Spent Reviewing:**

10 hours

---

> ### Author Response · Authors · 2021-08-10
> **Comments on abbreviations and structured variable examples**
>
> *“I didn't find what's the meaning of "SST", which is treated as the baseline in all the experiments.”*
>
> Sadly, this detail slipped out of our sight. The abbreviation stands for “Stochastic Softmax Tricks” [1], which is a relaxation-based gradient estimation framework for models with structured latent variables.
>
> *“I couldn’t understand what is “stochastic invariant”*
>
> We use the term stochastic invariant to refer to a specific property of Algorithm 2. In particular, if the input $E$ of the algorithm is a set of independent exponential random variables, then the input $E’$ of the recursive call to F_{struct} is also a set of independent exponential random variables (conditioned on the variables $T_1, …, T_m$ obtained above). Algorithm 2 has this property by design.
>
> Besides line 132, we try to explain the concept intuition for the top-k example on line 116.
>
> Later in the paper, we say that an algorithm has a stochastic invariant if it can be represented as an instance of Algorithm 2. We represent argtop-k as an instance of Algorithm 2 in Figure 2. In the supplementary material, we represent other algorithms as instances of Algorithm 2.
>
> *“Move the definition of execution trace, I could not understand table 1”*
>
> Thank you for the suggestion. We will modify the Section 3 structure to improve the narration and facilitate the perception of Table 1 and the stochastic invariant concept.
>
> *"The proposed method needs a lot of computation, however, it has larger variance than the standard X space score function estimator."*
>
> We want to emphasize that the processing time for a single batch in our framework does not significantly differ from the batch processing time of our baseline (see the answer to reviewer NYB7). On the other hand, obtaining the desirable standard score function involves the summation of a combinatorial number of terms.
>
> *“Could the author give a few practical examples where the score function is not available?”*
>
> A prominent example is the Gibbs distribution for matching [2, 3]. In this case, the probability of observing a permutation matrix $P$ is proportional to $exp(tr(P^t W))$ for a parameter matrix $W$. Obtaining a normalizing constant used in the score function is equivalent to computing a matrix permanent, which has #P computation complexity (i.e., at least as hard as NP).
>
> We also try to give a more intuitive example on lines 122-124. In the example, the score function for a fixed size subset is a sum of score functions for all ordered subsets. It contains $k!$ terms.
>
> [1] Paulus M. B. et al. Gradient Estimation with Stochastic Softmax Tricks //NeurIPS 2020. – 2020.
>
> [2] Taskar, B. (2004). Learning Structured Prediction Models: A Large Margin Approach. PhD thesis,
> Stanford University.
>
> [3] Valiant, L. G. (1979). The complexity of computing the permanent. Theor. Comput. Sci., 8(2):189–
> 201.

---

> > ### Comment · Reviewer_ahTk · 2021-08-13
> > **Thanks for clarification**
> >
> > Thanks to the author for the clarification, I do recognize the contribution of the paper so I raised my score, but my major concern is the writing which makes the paper hard to follow.
> >
> > My additional question is about the score function gradient. As the author mentioned, for some applications, the normalizer of the distribution is intractable to compute. However, we can still estimate the gradient up to a constant log normalizer, I am wondering will this gradient still give a useful optimization scheme?  Or Is it possible to show how it performs in the proposed situation? If that one doesn't work then the usefulness of the proposed can be reassured.

---

> > > ### Author Response · Authors · 2021-08-16
> > > **Comments on gradient estimates without the score function**
> > >
> > > Thank you for the feedback and the question. To adress it, we will consider a distribution $P_\theta(X=x) \propto \exp(-E_\theta(x))$ defined in terms of an energy function $E_\theta(x)$. The first example in our original answer is an instance of such distribution. The score function is the sum of the energy gradient and the gradient of the normalizing constant
> > > $$\nabla_\theta \log P_\theta(X=x) = -\nabla_\theta E_\theta(x) - \nabla_\theta \log Z(\theta) = -\nabla_\theta E_\theta(x) + \mathbb E_{X'} \nabla_\theta E_\theta(X').$$
> > >
> > > At first glance, using $-E_\theta(x)$ as a substitute for $\log P_\theta(X=x)$ does not result in a meaningful optimization scheme. In this case, the gradient is biased:
> > > $$\nabla_\theta \mathbb E_{X} f(X) = \mathbb E_{X} f(X) \nabla_\theta \log P_\theta(X) = \mathbb E_{X} f(X) \left[ - \nabla_\theta E_\theta(X) + \mathbb E_{X'} \nabla_\theta E_\theta(X') \right].$$
> > >
> > > Here the variable $X'$ has the same distribution as $X$. The bias is $\mathbb E_{X} f(X) \mathbb E_{X'} \nabla_\theta E_\theta(X')$ and is zero when $\mathbb E_X f(X) = 0$.
> > >
> > > After giving the question more thought, we note that an independent sample $X''$ from $P_\theta(X)$ eliminates the bias. The difference $g(X, X'') = f(X) - f(X'')$ gives an unbiased estimate for the original gradient:
> > > $$ \nabla_\theta \mathbb E_{X} f(X) = \mathbb E_{X, X''} g(X, X'') \nabla_\theta \log P_\theta(X).$$
> > >
> > > When we substitute the score function with $E_\theta(X)$, the bias is
> > > $$ \mathbb E_{X, X''} g(X, X'') \mathbb E_{X'} \nabla_\theta E_\theta(X') = 0$$
> > > because $\mathbb E_{X, X''} g(X, X'') = 0$. In general, we can obtain an unbiased estimate by replacing the score function with the energy gradient in REINFORCE+ estimator.
> > >
> > > Does this work well in practice? Obtaining an expression for the estimator is just half the story. We also need an efficient sampling algorithm to generate $x$ and $x''$ used in the estimate. An off-the-shelf Gibbs sampler may be too slow in this setup because we need to construct the gradients rapidly. Given the gradient estimate proposed above, adjusting a general sampling algorithm to SGD for such a model may be a fruitful topic for future work.
> > >
> > >
> > > In contrast with the above example, our work defines distributions implicitly based on sampling algorithms. Such distributions are more common within probabilistic deep learning as they provide an efficient sampling algorithm required for the forward pass and the gradient estimate. As a result, we do not have access to the energy function (i.e., log-probability up to a normalizing constant) and cannot employ the estimate described above. Instead, we highlight the trace score function $\nabla_\theta \log P_\theta(T=t)$ s.t.
> > > $$\nabla_\theta \log P_\theta(X=x) = \sum_{t: X(t) = x} \nabla_\theta \log P_\theta(T=t)$$
> > > and show that the trace score function gives a meaningful optimization scheme. The second example from our original answer illustrates such a setup with an ordered set score function used as a substitute for an unordered set score function.
> > >  In general, all the examples we consider in the paper have an intractable score function for the latent variable $X$ and a tractable score function for the trace $T$.

---

> > > > ### Comment · Reviewer_ahTk · 2021-08-17
> > > > **The explaination makes sense**
> > > >
> > > > I agree with the author that maybe for some distributions, the sampling might be a bottleneck and then the proposed method is useful.
> > > >
> > > > Therefore, I raised my score to 6. I encourage the authors to 1. make the paper easier to follow and  2.add the discussion about the score gradient when sampling is not easy, in the camera-ready version, if the paper is accepted.
> > > >
> > > > Thanks for the work during the rebuttal!

---

> > > > > ### Author Response · Authors · 2021-08-17
> > > > > **We will incorporate the discussion in the paper**
> > > > >
> > > > > Thank you for the suggestions and the prompt response. We will make sure to include the discussion of other distribution families in the future paper revision and will do our best to incorporate the feedback to improve the presentation.

---

### Official Review · Reviewer_NYB7 · 2021-07-16

**Rating:** 7
**Confidence:** 2

**Summary:**

This paper shows that, contrary to popular belief, score function gradient estimators for structured models are competitive to estimators based on continuous relaxations. The paper is based on the Gumbel-Max / competing exponential clock trick for structured models, and identifies that we can estimate gradients wrt the “trace” T of the program involved in the implicit mapping from the exponential clocks E to the observed X. This is most helpful when we can easily compute p(T), but not p(X).

**Limitations And Societal Impact:**

* The speed limitations due to the difficulty of batching algorithm 2 was made clear in the text.
* However, the limitations of the conditions of the algorithm itself are not as clear. While it was mentioned that Prim's algorithm cannot be handled, a better description of the limitations is needed.

**Main Review:**

*Originality*
* This work integrates tricks for gradient estimation using REINFORCE in discrete models and extends them to more complex settings. The use of REINFORCE on traces seems novel, and its success relative to continuous relaxations is surprising.

*Quality*
* The submission appears to be technically sound, and the experiments do support the claim that REINFORCE can achieve success comparable to continuous relaxations in structured settings. However, the numbers reported for the baseline SST do seem to sometimes be quite a bit lower than those reported in Paulus et. al. [1], which is concerning. For example, in Table 2 the complex model for SST has a mean MSE of 2.93, while all the complex models in Paulus et. al. [1] have mean MSE's <= 2.7.
* The experiment with non-monotonic generation of balanced parentheses  demonstrates that the proposed estimator is indeed more general than continuous relaxation-based methods.
* An analysis of the different gradient estimator’s MSEs compared to a continuous relaxation on a tractable structured model (so you can compute the exact gradient) would also be nice to see.

*Clarity*
* The submission could be improved by better motivating the problem and rationally reconstructing the proposed solution. One possible way of doing this could be to start by showing how gradient estimation in structured distributions is hard. Explain why, i.e. computing p(X) can be expensive, and give an example structured distribution that shows this. Use this to motivate the trace as a transformation T(E) = (X, Y), where although we have unused information Y, it is easier to compute p(T).
* The details on Gumbel-softmax following Lemma 1 do not seem relevant to the contribution of the paper (getting REINFORCE working) and can be removed.
* The details on variance reduction should come after the introduction of the trace-based estimator.
* The motivation for f_split in algorithm 2 should be in the main text, and required for the example.

*Significance*
* The results are important, and provide evidence that REINFORCE-based gradient estimators can be competitive with continuous relaxations.

*Questions*
* Why was stochastic invariant chosen as the keyword for the approach? Why not something like trace-based gradient estimators?
* Does the T-REINFORCE+ estimator only use the LOO estimator, and no continuous relaxation? If so, does this mean that the control variates that use conditional reparameterization never outperformed the LOO estimator?
* Instead of using a batch size of N/K and K samples, can the baseline SST be run with K noise samples in order to keep the batch size the same?
* What are the speed numbers (time per iteration)? Since batching was difficult, it is clear these will not be great, but it would still be good to show them.
* Can you detail exactly where the algorithm fails to apply to Prim's?

*Overall*
* Nice contribution, but experiments and writing need to be improved

[1] Max B Paulus, Dami Choi, Daniel Tarlow, Andreas Krause, and Chris J Maddison. Gradient estimation with stochastic softmax tricks. arXiv preprint arXiv:2006.08063, 2020.

Edit: Author response clarified experimental numbers. The remaining concern is writing, but the paper is understandable.

**Time Spent Reviewing:**

4

---

> ### Author Response · Authors · 2021-08-10
> **Comments on reported numbers, exposition and computation time**
>
> *"However, the numbers reported for the baseline SST do seem to sometimes be quite a bit lower than those reported in Paulus et. al. [1], which is concerning."*
>
> Compared to Paulus et al. [1], we have a different experimental protocol. They report the best MSE and standard deviations across sampled hyperparameters for a fixed model initialization (see [1, Appendix D.3] for further details). We optimize the hyperparameters using the random search for fixed model initialization. Then we report the mean performance across different random initializations for the best hyperparameter set. Although both protocols are viable, the second is more common for measuring the model performance. Still, we checked that our implementation achieves the numbers reported for SST.
>
> *"The details on Gumbel-softmax following Lemma 1 do not seem relevant to the contribution of the paper (getting REINFORCE working) and can be removed."*
>
> We respectfully disagree on this point. Gumbel-softmax plays a role in motivating the vanilla RELAX estimator and our baseline. We consider getting the extension of RELAX working to be another valuable contribution despite the underwhelming performance. We provide these details to introduce RELAX and keep the manuscript self-contained.
>
> *" Why was stochastic invariant chosen as the keyword for the approach? Why not something like trace-based gradient estimators? "*
>
> The word "trace" is a widely used term that applies to any algorithm. At the same time, we can compute the trace score function only for a limited set of algorithms. Our motivation was to use the term "stochastic invariant" to distinguish the specific algorithm family for which we derive the estimators.
>
> *" Does the T-REINFORCE+ estimator only use the LOO estimator and no continuous relaxation? If so, does this mean that the control variates that use conditional reparameterization never outperformed the LOO estimator? "*
>
> Indeed T-REINFORCE+ and E-REINFORCE+ only use the LOO estimator without any continuous relaxations. And T-REINFORCE+ performs better than RELAX estimator with sample-dependent baseline. Recently, [2, 3] showed that the LOO estimator is indeed a strong baseline. Thus the experimental results provided in our paper can also support this claim. Concerning the worse performance of the RELAX estimator, [4] argued that despite the popular belief sample-dependent baselines may not have the best performance. Still, RELAX may be preferable in the setups where multiple samples required for LOO are not available (i.e., reinforcement learning).
>
> *" Instead of using a batch size of N/K and K samples, can the baseline SST be run with K noise samples in order to keep the batch size the same? "*
>
> Certainly, SST can be run with K samples; however, reparameterization-based gradient estimates (including SST) typically use a single sample. The variance of SST is already low and $K$ additional samples will only decrease the variance proportionally to $K$ ($K$ is small). On the other hand, the large batch size is often desirable for neural net training. Therefore, we expect using $K$ noise sample to give a smaller advantage to SST than using a reduced batch in T-REINFORCE+.
>
> *" What are the speed numbers (time per iteration)? Since batching was difficult, it is clear these will not be great, but it would still be good to show them. "*
>
> Below you can find the average processing time for one batch in Graph Layout (Spanning Tree) and ListOps (Arborescence) experiments.
>
> | Time per iter (ms) | SST | T-REINFORCE+ | RELAX |
> |:---:|:---:|:---:|:---:|
> | Spanning Tree | 123 | 137 | 180 |
> | Arborescence | 175 | 249 | 535 |
>
> Implementation of Top-K is fully vectorized and supports efficient batching. Implementation of Spanning Tree is vectorized for batch dimension but involves a for loop with a number of steps equal to a number of vertices. Therefore, one will expect greater speed reduction with an increase in the number of vertices rather than an increase in the batch size. We implement algorithms for Arborescence in C++. In this case, sampling X, E | T, and the likelihood computation involve a loop over the batch elements. Although the computation of T-REINFORCE+ and RELAX gradient estimators are slower, they are still applicable in practice.
>
> *" Can you detail exactly where the algorithm fails to apply to Prim's?"*
>
> Both Kruskal's and Prim's algorithms are greedy algorithms for finding minimum spanning trees. However, they select the minimum edge from different edge sets. Kruskal's algorithm starts by considering all edges and gradually excludes edges from the consideration after every step. We use this property to rewrite Kruskal's algorithm as an instance of Algorithm 2. Conversely, Prim's algorithm finds the minimum edge among the edges incident to a partially constructed tree. The distributions of the remaining edge weights conditioned on the selected edge are not independent exponential. As a result, we cannot represent the algorithm as an instance of Algorithm 2.
>
> For example, consider a graph with 3 vertices and 3 edges with weights $E_{12}, E_{13}, E_{23}$. Prim's algorithm compares $E_{12}$ and $E_{13}$ and adds edge $(1,2)$ on the first step. Conditioned on the selected edge, the distribution of $E_{23}$ remains unchanged, whereas $E_{13}$ becomes an exponential distribution truncated at the minimum value $E_{12}$. The exponential-min trick will not apply in this case.
>
> In the same setup, Kruskal's algorithm will find the global minimum edge on the first step, for example, edge $(1, 2)$. The distributions of $E_{23}$ and $E_{13}$ conditioned on the selected edge will be exponential distributions truncated at $E_{12}$. Subtracting $E_{12}$ leads to two exponential distributions and does not change the minimum spanning tree.
>
> [1] Max B Paulus, Dami Choi, Daniel Tarlow, Andreas Krause, and Chris J Maddison. Gradient estimation with stochastic softmax tricks. NeurIPS 2020.
>
> [2] Lorenz Richter, Ayman Boustati, Nikolas Nüsken, Francisco J. R. Ruiz, Ömer Deniz Akyildiz. VarGrad: A Low-Variance Gradient Estimator for Variational Inference. NeurIPS 2020.
>
> [3] Zhe Dong, Andriy Mnih, George Tucker. Coupled Gradient Estimators for Discrete Latent Variables. AABI 2021 Symposium
>
> [4] George Tucker, Surya Bhupatiraju, Shixiang Gu, Richard E. Turner, Zoubin Ghahramani, Sergey Levine. The Mirage of Action-Dependent Baselines in Reinforcement Learning. ICML 2018
>
> [5] Max B. Paulus, Chris J. Maddison, Andreas Krause. Rao-Blackwellizing the Straight-Through Gumbel-Softmax Gradient Estimator. ICLR 2021

---

### Official Review · Reviewer_npTm · 2021-07-16

**Rating:** 5
**Confidence:** 4

**Summary:**

Mathematically, this paper proposes a framework for designing a structured latent variable X and also for training the related objective E_{X} [L(X)].

The structured latent variable X could model ranking, subsets, sequences, spanning trees, rooted trees, etc.

The framework implicitly models X with Exponential random variables followed by a recursive calculation; the corresponding P(X; \lambda) could be intractable. Here, after one step of the calculation, the resulting output will also be Exponentially distributed; the authors name such phenomenon (unchanged Exponential distribution) as "Stochastic Invariant."

To train the objective E_{X} [L(X)], where P(X; \lambda) is intractable, the authors propose to do REINFORCE with internal variable T from the above recursive calculation.

Experiments on a variety of situations are conducted to evaluate the proposed framework.

**Limitations And Societal Impact:**

Yes.

**Main Review:**

Originality. The presented framework is believed novel. However, the relationship between the proposed framework and related work (such as those mentioned in Section 3.3) is not clear. The detailed discussions/comparisons are necessary to highlight the novelties here. For example, the proposed framework seems to include Kruskal's algorithm [21] and Chu-Liu-Edmonds [6] as special cases, right?

Quality. The presented framework is technically sound. The theoretical analysis and experimental results are convincing.

Clarity. The writing is clearly not friendly to general machine learning audiences. I have to read the paper several times to understand the main idea. General audiences would appreciate simple and straight statements of the problem settings and contributions.
Definitions should be presented early. For example, what are structured latent variables, Stochastic Invariant, etc? For example, I don't understand the term of Stochastic Invariant until Line 116. Besides, these definitions are much easier to understand if their mathematical expressions are provided.

Significance. The presented framework is likely to be important and to be reused by other researchers.

Other comments.

When I read the submission for the first time, I am completely lost after seeing Table 1. Honestly, this table, placed here, is not easy to understand at all.

Notations could be improved. For example, use a bold symbol to denote a vector; replace X^{-1} in Line 124 with another notation.

In Lines 135-145, the sub-routines, e.g., f_{map}, are not defined.

Line 152. Lemma "2" is a typo.

The authors seem to have considered many extensions of their framework, for example, in Section 3.3 and the appendix. However, that is not clearly stated in the current submission. I would suggest leveraging a table to summarize the extensions and using mathematical definitions for clear demonstrations.


**Time Spent Reviewing:**

8

---

> ### Author Response · Authors · 2021-08-10
> **Comments on the related work and formal definitions**
>
> *"However, the relationship between the proposed framework and related work (such as those mentioned in Section 3.3) is not clear. The detailed discussions/comparisons are necessary to highlight the novelties here. For example, the proposed framework seems to include Kruskal's algorithm [21] and Chu-Liu-Edmonds [6] as special cases, right?"*
>
> The general Algorithm 2 and the $T$-score estimator are the main contributions of the paper. The algorithm acts as a unifying paradigm. It includes all the examples mentioned in Section 3.3 (lines 160-161), including Kruskal and Chu-Liu-Edmonds algorithms. In Section 5 (Related Work), we highlight that the Gumbel Top-K (lines 224-228), Kruskal and Edmonds algorithms (lines 228-232) with Gumbel/Exponential inputs occurred in the literature before. However, we emphasize that the proposed general training paradigm allows constructing a training algorithm for these particular examples (line 231).
>
> Another two stochastic invariant algorithms that we introduce in Section 3.3 produce matchings and binary trees, respectively. To the best of our knowledge, such algorithms and the resulting distributions are novel.
>
> *"Definitions should be presented early. For example, what are structured latent variables, Stochastic Invariant, etc? For example, I don't understand the term of Stochastic Invariant until Line 116. Besides, these definitions are much easier to understand if their mathematical expressions are provided."*
>
> We discuss structured latent variables in lines 29-40 of the introduction and provide motivation and limitations of their use. "Structured" in structured prediction is an umbrella term for a wide variety of variables. We did not encounter a formal definition in the literature, so we tried to introduce the notion based on examples and their properties (lines 29-40).
>
> The term Stochastic Invariant is new. We first explained in detail for an illustrative example on line 116 at the beginning of the main section. Before the main section, we introduced related concepts (the exponential-min trick and various related gradient estimators) to make the draft accessible to general machine learning audiences.
>
> In general, we tried to avoid overcrowded definitions and explain the framework through examples. We will try to introduce more formal notation in Section 3 to support the exposition.
>
> *"When I read the submission for the first time, I am completely lost after seeing Table 1. Honestly, this table, placed here, is not easy to understand at all."*
>
> We will reorganize the text and the position of the table in a way to better coordinate with the analysis of gradient estimation in Section 4.\\
>
> *"In Lines 135-145, the sub-routines, e.g., $f_{\text{map}}$, are not defined."*
>
> Indeed, thank you. In Algorithm 2, we provide a general template of an algorithm. The sub-routines may vary, which allows implementing different algorithms. On lines 135-145, we refer to the sub-routines used in Algorithm 2 and explain the motivation behind the template design.

---

### Official Review · Reviewer_SiHy · 2021-07-17

**Rating:** 7
**Confidence:** 3

**Summary:**

Endowing a latent representation with particular structural properties is a common way to incorporate inductive biases in deep learning. Structured latent variable models often require discrete components which can make optimisation of these models challenging. Typically, learning approaches use (i) differentiable surrogates which rely on relaxations (to continuous components) and thereby introduce additional constraints and biases in gradient estimation or (ii) score function estimators which lead to high variance in the gradient estimates. This submission proposes a framework providing low-variance score function estimators on the basis of a property termed *stochastic invariance* and an intermediate *trace* variable $T$. The invariance describes that the distribution of the perturbed (exponential) inputs $E$ does not change in the proposed recursive algorithms, i.e. stays invariant throughout the recursion. Empirical evaluation is performed on different common tasks with structured latent variables: (i) identifying fixed-size subsets of predictors which are most informative for target prediction (*BeerAdvocate* dataset), (ii) graph-represented particle interactions (*Graph Layout*), (iii) recovering of parse trees / arborescence (simplified *ListOps* dataset), and (iv) a generated dataset of parentheses (akin to language models) where relaxation-based approaches might not be feasible. The considered models are based on CNNs, GNNs, and LSTMs arranged in encoder-decoder or encoder-classifier architectures. In particular, models employing stochastic softmax tricks (SST) and a relaxed version of the used score function estimator are compared to the proposed method variants. The results indicate that the proposed approach can generally compete well or even outperform these baselines.

**Limitations And Societal Impact:**

Yes, the authors have adequately addressed the limitations and societal impact, in my opinion (see page 9 lines 326-329).

**Main Review:**

## 1. Strengths

### Originality / Novelty:
- This submission derives conditions and algorithms for which score function estimators (REINFORCE) can yield low-variance gradient estimates and compete with SOTA relaxation-based approaches to learning of structured latent variables. In this work, the perturbed (exponential) input $E$ is connected via an intermediate trace variable $T\equiv T(E)$ to structured (target) variable $X\equiv X(T)$. The proposed recursive algorithms make use of stochastic invariance which takes advantage of the Gumbel-Max trick. The key aspect of this work is that computations can be made w.r.t. trace $T$ and compute score functions on the basis of $T$ ($T$-REINFORCE+) with reduced variance.
- The underlying idea of stochastic invariance and trace $T$ is not novel and the submission shares similarity to other work which was a main discussion point in a previous submission to ICML 2021. However, in my opinion this submission convincingly highlights the differences to related work and puts emphasis on practical usage of $T$ for score function estimation and conditional reparameterisation. As such I believe the work extends the current literature and yields some novel aspects.

### Clarity:
- I consider this submission well-structured, clearly presented and mostly self-contained.

### Quality:
- The empirical as well as formal and more technical aspects of this submission appear sound as far as I can judge.
- In my opinion, the submission succeeds in demonstrating that the proposed approach indeed provides empirical advantages. It is shown that the $T$-REINFORCE+ approach generally outperforms $E$-REINFORCE+ and competes mostly well with the relaxation-based models.

### Significance / Relevance:
- The submission extends the literature concerning score function estimators in the above described ways. This work might contribute to making score function estimators more applicable and relevant for a larger audience.

## 2. Weaknesses

### Clarity:
- Details on auxiliary variable $R$: I am unsure I fully understand the meaning and necessity of auxiliary variable $R$. It is stated that it *”[…] accumulate[s] the necessary information for the next call […]”* (lines 141-142), but I do not quite follow here. This is probably a more technical detail, but could the authors give some details on that?
- *”Notably, the multi-sample estimator with a smaller batch is more stable than the single-sample estimator with a sample-dependent control variate.”* (lines 321-322) I am not sure I can follow in this conclusion here. Where was this shown or what are the authors referring to here?

### Quality:
- One goal of the research here is to *”[…] allow learning models that do not admit relaxed variables […]”* (l. 45) which is attempted to be addressed in the last experiment (section 6.4). As I have more limited experience in this direction, could the authors maybe comment on how relevant this task is. Is there for instance a class of problems where it is clear that relaxation-based approaches are not viable? In the end, it might be a less pressing matter as apparently the community also moved more towards relaxation-based approaches and identifying a surrogate formulation which is differentiable. I believe the submission would greatly benefit from such an exposition.

## 3. Additional Feedback

Some minor aspects the authors might want to take into consideration when revising the submission:

- Figure 1, Algo.s 1 & 2: I might be mistaken, but should it not read $T\Leftarrow$ argmin $E$ and $T_i\Leftarrow$ argmin $E_i$ in these algorithms? It could have been meant to highlight what is done there more explicitly, but I rather stumbled upon that.
- Lines 111-113: I might have missed the explanation to which is referred here, but I could not follow why $E’_j=E_j-E_T$ might be required / beneficial. Could the authors point out where this is explained or comment on that?
- L. 128, missing dot at the end of the sentence.
- L. 152: “[…] Lemma 2 […]” should probably be “[…] Lemma 1 […]”
- L. 202: *“As we’ve argued […]”* -> “As we have argued […]”
- SST abbreviation: It might be advisable for clarity to introduce the abbreviation of stochastic softmax tricks (SST) in the main text (currently in the appendix, L. 666).


## 4. Recommendation

I believe this is a well-written and self-contained submission which contributes new insights to score function estimators and learning structured latent variables. In my opinion the work highlights differences to other related papers and extends these. Therefore, I consider this submission a worthy contribution to NeurIPS and my initial recommendation is to accept this paper.

### Post-Rebuttal Update:
I would like to thank the authors for their thorough rebuttal and additional clarifications. I stand with my initial assessment and endorse accepting this submission for publication at NeurIPS.



**Time Spent Reviewing:**

10

---

> ### Author Response · Authors · 2021-08-10
> **Comments on related areas and technical details**
>
> *"Is there for instance a class of problems where it is clear that relaxation-based approaches are not viable? In the end, it might be a less pressing matter as apparently the community also moved more towards relaxation-based approaches and identifying a surrogate formulation which is differentiable. I believe the submission would greatly benefit from such an exposition."*
>
> One of the fields is neural program synthesis [1, 2], which combines neural networks with program learning. Such models introduce prior knowledge about the structure of data and may make NNs more interpretable. They usually contain a block that generates a program that needs to be executed to obtain the output. If we do not have access to the ground truth programs (which is often the case in practice), the program becomes a discrete latent variable. It seemingly does not admit continuous relaxations since the program needs to be executed. In this way, we need to use gradient estimation techniques for training. Since programs typically have syntactic structures, methods for tree-like structures can facilitate generating a syntactically correct output.
>
> Another wide application area is reinforcement learning (RL) and related problems like simulation-based inference, experimental design, etc. In RL, the goal is to learn policy distribution to select actions leading to maximal rewards. In many RL setups, action space is naturally discrete - at each step, you have to make a single decision. Either common issue is that the reward function is black-box, meaning that you only have access to the function itself, but not its gradient. This issue prevents using gradient estimation techniques based on continuous relaxations. For further review on differences and applicability of different gradient estimators see [3].
>
> [1] Yi, K., Wu, J., Gan, C., Torralba, A., Kohli, P., & Tenenbaum, J. (2018, January). Neural-Symbolic VQA: Disentangling Reasoning from Vision and Language Understanding. In NeurIPS.
> [2] Mao, J., Gan, C., Kohli, P., Tenenbaum, J. B., & Wu, J. (2019). The neuro-symbolic concept learner: Interpreting scenes, words, and sentences from natural supervision. arXiv preprint arXiv:1904.12584.
> [3] Shakir Mohamed, Mihaela Rosca, Michael Figurnov, Andriy Mnih. Monte Carlo Gradient Estimation in Machine Learning. Journal of Machine Learning Research.
>
> *"Details on auxiliary variable R. I am unsure I fully understand the meaning and necessity of auxiliary variable R. It is stated that it ”[…] accumulate[s] the necessary information for the next call […]” (lines 141-142), but I do not quite follow here. This is probably a more technical detail, but could the authors give some details on that?"*
>
> The main goal of introducing the auxiliary variable R is to make Algorithm 2 general. Almost all algorithms need to maintain some information about the current state of execution. Maintaining just the set of indices (K) and the corresponding values (E) can be not enough. This information and intermediate transformations (formalized in out terms as functions $f_{\text{stop}}$, $f_{\text{map}}$ and $f_{\text{combine}}$) vary in different algorithms, but their particular form does not affect the stochastic invariance, if the overall procedure is represented in the form of Algorithm 2. Given this, we do not specify the variety of possible operations in the algorithms. This ensures that the family of stochastic invariant algorithms is large enough to cover algorithms for different discrete structures.
>
> For example, in the Top-k algorithm (Alg. 1), described as sequentially taking indices of the $k$ smallest array elements, we need to know, whether the current subset size is equal to $k$. We put this information into the variable $R$ since it cannot be extracted from $K$ and $E$.
>
> In Kruskal's algorithm (Alg. 10) for minimum spanning trees, $R$ corresponds to the disjoint set union data structure. It efficiently maintains the set of acceptable edges that do not produce cycles and, thus, are suitable for the spanning tree. We cannot store such information in $K$ and $E$; therefore, we use the auxiliary variable $R$.
>
> Algorithms 7-11 in Appendix F present other examples.
>
> *”Notably, the multi-sample estimator with a smaller batch is more stable than the single-sample estimator with a sample-dependent control variate.” (lines 321-322) I am not sure I can follow in this conclusion here. Where was this shown or what are the authors referring to here?"*
>
> Here we refer to Tables 2, 3, 4, which show that the performance of T-REINFORCE+ is almost always better than the performance of RELAX. In these experiments, we fix the number of loss function evaluations $N$. It is equal to the batch size for SST and RELAX. However, T-REINFORCE+ uses $K$ latent samples and loss function evaluations per object; we reduce the batch size to $N/K$ to facilitate fair comparison. Currently, a description of the detail about batch size is placed in Appendix E, lines 666-670.
>
> *"Figure 1, Algo.s 1 and 2: I might be mistaken, but should it not read $T = \arg\min E$ and $T_i = \arg\min E_i$ in these algorithms? It could have been meant to highlight what is done there more explicitly, but I rather stumbled upon that."*
>
> In both cases, $E$ is a vector and $E_j$ is the value of its $j$-th coordinate. We work with sets of indices only through index set $K$ or its partition $\{P_i\}$. In the first case with Top-k we take the index $j$, minimizing $E_j$ among all $j \in K$. In the second case, we make the same thing for each $i$ considering indices $j \in P_i$.
>
> *"Lines 111-113: I might have missed the explanation to which is referred here, but I could not follow why $E'_j = E_j - E_T$ might be required / beneficial. Could the authors point out where this is explained or comment on that?"*
>
> This step is a technical detail simplifying the analysis. Updating $E$ into $E'$ is necessary for maintaining the exponential distribution after each step and originates from the exponential-min-trick (Appendix B1, Lemma 2). Most of the algorithms considered in the paper would return the same results without this update.

---

> > ### Comment · Reviewer_SiHy · 2021-08-24
> > **Questions answered**
> >
> > I would like to thank the authors for the clarifications. Considering also the remarks of the other reviewers, it appears that the submission would benefit greatly from adding some more details and explanations to the main text. These could include:
> > - a clearer motivation (see also first point in *clarity* of rev. *NYB7*) similar to the first answer (relevant problems with no relaxation applicable)
> > - a (short) statement or example on auxiliary variables and functions $R$, $f_\text{split}$, $f_\text{map}$ etc. (see also rev.s *NYB7* and *npTm*) to give an idea how these can be realised, again similar to the answer provided in the response
> > - a clearer statement what is meant by *stability* in the conclusions, i.e. the reply concerning improved performance of T-REINFORCE+ compared to RELAX
> >
> > I believe these adjustments can be made with a few additional sentences requiring only a minor revision.

---

> > > ### Author Response · Authors · 2021-08-27
> > > **Thanks to the reviewer**
> > >
> > > We would like to thank the reviewer for the valuable comments and suggestions given in the review and during rebuttal period. We will revise our draft accordingly to include the clarifications about auxiliary variables and functions as well as the clearer exposition of the proposed method.

---

### Author Response · Authors · 2021-08-10
**A general note**

We would like to thank the reviewers for their time spent providing us with a detailed evaluation of our work and fruitful suggestions. The general concerns regard the lack of clarity and polish of the manuscript. We will make our best effort to take into consideration all of the suggestions and improve the narration in our paper. Below we personally address reviewers’ individual concerns.

---

### Decision · Program_Chairs · 2021-09-28

**Decision:**

Accept (Poster)

**Comment:**

This paper exploits a recent observation that exponential random variables, when filtered through an algorithm that makes local, greedy choices, can sometimes keep their distributions invariant. The paper shows that in such cases, this invariance can be exploited to provide lower variance REINFORCE gradient estimators. After the response period and discussion, most reviewers agreed that this is an interesting and valuable contribution. Some reviewers felt that the lack of clarity in the writing significantly diminished the contribution, but no reviewers felt that this disqualified the paper from publication.

I am convinced that this paper merits publication for two reasons.
* I believe the stochastic invariant structure is of independent interest in cases beyond those considered here.
* The insight that stochastic invariants allow for lower variance gradient estimation is a valuable one.

For the authors to achieve the maximum impact possible with this paper, I suggest that they take seriously the feedback on clarity. In particular, that they consider *minor* revisions to address the motivation, definition of stochastic invariance, and details of the algorithm.

**Consistency Experiment:**

NeurIPS has a long history of experimentation. In 2014, NeurIPS ran an experiment in which 10% of submissions were reviewed by two independent committees to quantify the randomness in the review process. This year, we repeated a variant of this experiment to see how the quality of the review process has changed over time.  This paper was part of the experiment and was therefore assigned to two committees (consisting of reviewers, an Area Chair, and a Senior Area Chair) that reached independent decisions.  If both committees made the same recommendation, this recommendation was followed. If a single committee recommended acceptance, the paper was accepted (with the exception of a few cases in which the other committee identified what we considered a fatal flaw, e.g., an error in a key result).

Both committees reached the same decision: **Accept (Poster)**

The other committee assigned to the paper recommended **Accept (Poster)**.  You can find the other set of reviews, along with any follow up discussion with the authors here:
https://openreview.net/forum?id=V2nQ_he-go_